# A novel biosensor for the spatiotemporal analysis of STING activation during innate immune responses to dsDNA

Steve Smarduch [ID] [1,8], Sergio David Moreno-Velasquez [ID] [1,8], Doroteja Ilic[2], Shashank Dadsena [ID] [3], Ryan Morant [ID] [1], Anja Ciprinidis[1], Gislene Pereira [ID] [1,4], Marco Binder[2], Ana J García-Sáez [ID] [3,5] & Sergio P Acebrón [ID] [1,6,7 ✉]

## Abstract

**The cGAS-STING signalling pathway has a central role in the innate immune response to extrinsic and intrinsic sources of cytoplasmic dsDNA. At the core of this pathway is cGAS-dependent production of the intra- and extra-cellular messenger cGAMP, which activates STING and leads to IRF3-dependent expression of cytokines and interferons. Despite its relevance to viral and bacterial infections, cell death, and genome instability, the lack of specific live-cell reporters has precluded spatiotemporal analyses of cGAS-STING signalling. Here, we generate a fluorescent biosensor termed SIRF (STING-IRF3), which reports on the functional interaction between activated STING and IRF3 at the Golgi. We show that cells harbouring SIRF react in a time- and concentration-dependent manner both to STING agonists and to microenvironmental cGAMP. We demonstrate that the new biosensor is suitable for single-cell characterisation of immune responses to HSV-1 infection, mtDNA release upon apoptosis, or other sources of cytoplasmic dsDNA. Furthermore, our results indicate that STING signalling is not activated by ruptured micronuclei, suggesting that other cytosolic pattern recognition receptors underlie the interferon responses to chromosomal instability.**

**Keywords** cGAS-STING Signalling; cGAMP; Biosensor; Micronuclei; Innate Immune Response
**Subject Categories** Immunology; Methods & Resources

## Introduction

The cGAS-STING signalling pathway has been proposed as a versatile and unifying sensing mechanism for double-stranded DNA (dsDNA) that is coupled to inflammation in mammals (Decout et al, 2021; Ishikawa et al, 2009; Margolis et al, 2017). The catalytic activity of cGAS is stimulated by double-stranded DNA (dsDNA) (Ablasser et al, 2013a; Civril et al, 2013; Sun et al, 2013) from any source, and therefore uncoupled from pathogen- or host-specific features (Decout et al, 2021). Mechanistically, a positive charged nucleotidyl-transferase domain binds the dsDNA backbone thereby inducing allosteric changes that favour ATP and GTP binding, and the subsequent synthesis of 2′3′-cyclic GMP–AMP (cGAMP) (Civril et al, 2013). Dimerization of cGAS results in a 2:2 DNA-cGAS complex and a ladder-like structure along the dsDNA backbones (Andreeva et al, 2017; Luecke et al, 2017; Shu et al, 2014). In vitro analyses demonstrated that longer dsDNA fragments (over 4 kbp) are more efficient activators of cGAS, possibly due to emerging properties associated to phase-separation (Du and Chen, 2018; Luecke et al, 2017).

STING is an endoplasmic reticulum (ER) resident transmembrane protein. STING consists of a lumen N-terminal domain, 4-span transmembrane domains, and a cGAMP ligand binding domain facing the cytoplasm and bound to a C-terminal tail (Shang et al, 2019). In the absence of cGAMP, STING forms homodimers with the ligand binding domains interlinked and covered by their respective C-terminal tails. STING-cGAMP complex formation induces allosteric changes that i) lead to additional oligomerisation between activated dimers via bisulphite bridges and ii) presentation of the C-terminal tails (Ergun et al, 2019; Huang et al, 2012; Ishikawa et al, 2009; Shang et al, 2019; Shang et al, 2012). Activated STING is trafficked to the Golgi via the COPII machinery in a process that is not yet well-understood (Gui et al, 2019; Mukai et al, 2016). At the Golgi, STING is palmitoylated, which is essential for its activity (Mukai et al, 2016). Upon activation, the C-terminal tails of the STING complexes recruit the TANK Binding Kinase 1 (TBK1), which dimerises through auto-phosphorylation. TBK1 also phosphorylates the C-terminal tails of STING (Zhang et al, 2019), which is required for the subsequent recruitment and phosphorylation of the interferon regulatory factor 3 (IRF3) (Liu et al, 2015). Phosphorylated IRF3 dimerises and

[1]Centre for Organismal Studies (COS), Heidelberg University, Heidelberg, Germany. [2]Division of Virus-associated Carcinogenesis, German Cancer Research Center (DKFZ), Heidelberg, Germany. [3]Institute of Genetics, CECAD, University of Cologne, Cologne, Germany. [4]Molecular Biology of Centrosome and Cilia, German Cancer Research Centre (DKFZ), Heidelberg, Germany. [5]Max Planck Institute of Biophysics, Frankfurt, Germany. [6]IKERBASQUE, Basque Foundation of Science, Bilbao, Spain. [7]University of the Basque Country (UPV/EHU), Leioa, Spain. [8]These authors contributed equally: Steve Smarduch, Sergio David Moreno-Velasquez. ✉E-mail: sergio.acebron@cos.uni-heidelberg.de

translocates to the nucleus where it induces the transcription of type I interferon genes, interferon-stimulated genes, and inflammatory cytokines (Doyle et al, 2002).

In the context of host DNA, both mitochondrial DNA (mtDNA) during apoptosis, as well as genomic DNA (gDNA) resulted from DNA damage or chromosome missegregation can induce cGAS-STING signalling. During apoptosis, the pro-death factors BAK and BAX oligomerise at the mitochondrial outer membrane (Cosentino et al, 2022), leading to its permeabilization and the release of cytochrome C and mtDNA to the cytoplasm (Cosentino et al, 2022; McArthur et al, 2018). Interestingly, cytochrome C signalling leads to caspase activation that can attenuate cGAS-STING signalling by cleaving of cGAS, TBK1 and IRF3 to ensure that the immunogenic response is not activated during apoptosis (White et al, 2014). In the absence of caspase-signalling, the efflux of mtDNA fully activates cGAS-STING signalling (Cosentino et al, 2022; Giampazolias et al, 2017; White et al, 2014). Missegregated chromosomes, broken gDNA fragments arisen from chromosome bridges, or otherwise damaged DNA, are normally encapsulated by their nuclear envelope in the daughter cells in so-called micronuclei (Harding et al, 2017; Zhang et al, 2015). Damaged DNA at the micronuclei undergoes chromothripsis (extreme rearrangements and pulverisation) due to the collapse of the lamina-defective nuclear envelope (Zhang et al, 2015). This process exposes the DNA to the cytoplasm leading to the accumulation of GMP–AMP synthase (cGAS) in the micronuclei and the subsequent activation of STING signalling (Harding et al, 2017; Mackenzie et al, 2017). Of note, an inhibitory phosphorylation by CDK1 and Aurora B (Davenport et al, 2016; Li et al, 2021), as well as membrane tethering (Li et al, 2021), prevents cGAS activation by condensed chromosomes during a normal mitosis, although it can allow slow accumulation of IRF3 and subsequent apoptosis upon mitotic arrest (Zierhut et al, 2019). On the other hand, chromatin bridges during mitosis have been shown to induce cGAS activation (Flynn et al, 2021). Intriguingly, structural studies indicate that cGAS can also be sequestered and inhibited by the nucleosome histones H2A/B (Michalski et al, 2020; Pathare et al, 2020), providing a rationale for preventing autoreactivity to the host genomic DNA.

In the context of foreign DNA, viruses, bacteria, as well as gDNA/mtDNA released from dead cells can trigger the activation of cGAS-STING. For instance, infection of macrophages by Herpes Simplex Virus 1 (HSV-1) leads to a quick response towards the viral DNA by cGAS (Reinert et al, 2016), which triggers interferon response necessary for the clearance of the infected cells. Similarly, intracellular replication of the gram-positive bacterium *Listeria monocytogenes* triggers cGAS-STING activation and interferon expression in myeloid cells (Hansen et al, 2014).

Intracellularly accumulated cGAMP can be exported by ABCC1 transporters to the microenvironment (Maltbaek et al, 2022), where it functions as immune-transmitter (Ablasser et al, 2013b) imported by SLC19A1 and SLC46A2 (Cordova et al, 2021; Ritchie et al, 2019), including in macrophages and monocytes. However, this immune response can be hampered by the ecto-nucleotide pyrophosphatase/phosphodiesterase ENPP1 (Carozza et al, 2020), which degrades extracellular cGAMP (Li et al, 2014). Further, cGAMP also spreads to bystander cells through gap-junctions, thereby promoting local immune response (Ablasser et al, 2013b; Chen et al, 2016).

Although several transcriptional reporters of cascades downstream of STING have been used in the past, including the INFb-Luciferase reporter, these approaches have a long latency on their response—often conflicting with cell death—lack spatio-temporal information, can be targeted by other signalling cascades, or rely on complex analyses (e.g., FRET) (Pollock et al, 2020). As such, the study of the innate immune response to dsDNA in live cells is hampered by the lack of robust biosensors and reporters. The STING receptor in particular, acts as a bottleneck in the surveillance of any source of cytoplasmic dsDNA. As such, engineering STING as a biosensor could provide an opportunity to monitor many different inputs ranging from chromosomal stability, viral/bacterial infections, apoptosis and other forms of cell death, auto-immunity and tumour immunity, including cancer therapy.

## Results

### Design and validation of a novel cGAS-STING-IRF3 biosensor

We turned our attention towards the mechanisms of STING activation and designed different chimeric proteins using cGAS, STING, TBK1, IRF3, NFκβ, and IKK, as well as linkers of various lengths, to couple their activation or interaction with luminescence, CRISPRa, and fluorescence reporters. Among the tested candidate constructs, we selected a tandem split GFP approach between STING and its downstream target IRF3, which we called SIRF (STING-IRF3) (Fig. 1A,B and [Expanded view] EV1A). In detail, we generated a construct containing STING-GFP11$_{x3}$ and IRF3-GFP1-10 separated by a ribosome skipping peptide (P2A) to facilitate stochiometric expression. To avoid the constitutive activation of STING signalling due to plasmid transfection of the SIRF biosensor and to study its function in different cell models, we generated HEK293T, HeLa, and human fibroblast stable cell lines using the PiggyBAC transposase system. Western blot and qRT-PCR analyses of HeLa cGAMP-biosensor cells showed that treatment with the STING agonist diABZI induced phosphorylation of STING-GFP11$_{x3}$ (Fig. 1C) and the endogenous downstream effector TBK1 (Fig. EV1B), as well as downstream *IFNB* (interferon β) expression (Fig. EV1C), confirming the functionality of the modified receptor.

Live cell imaging of the SIRF biosensor in HeLa cells showed a diffuse EGFP signal in the ER (Fig. 1D and Movie EV1). Treatment with the STING agonist diABZI (0.5 μM) led to quick clustering of the SIRF biosensor within ~1 h (Fig. 1D–F and Movie EV2), reaching a 4-fold increase in fluorescence at 8 h. Consistent with its requirement for STING-IRF3 interaction, inhibition of ER to Golgi transport via co-treatment with Brefeldin A, fully blocked the activation of the biosensor in HeLa cells (Fig. 1D,E). Of note, virtually all cells harbouring the biosensor (97%) responded to 0.5 μM diABZI within the first 4 h, while only 2% of the controls displayed activation of the biosensor (Fig. 1F and Movie EV2). Analyses of IRF3 nuclear localisation by immunofluorescence in wt HeLa cells confirmed that physiological response of the biosensor to diABZI and Brefeldin A (Fig. EV1D,E). Immunofluorescence analyses also confirmed the co-localisation of the inactive SIRF biosensor at the ER (Fig. 1G, calnexin). The activated biosensor localised in large vesicles around the

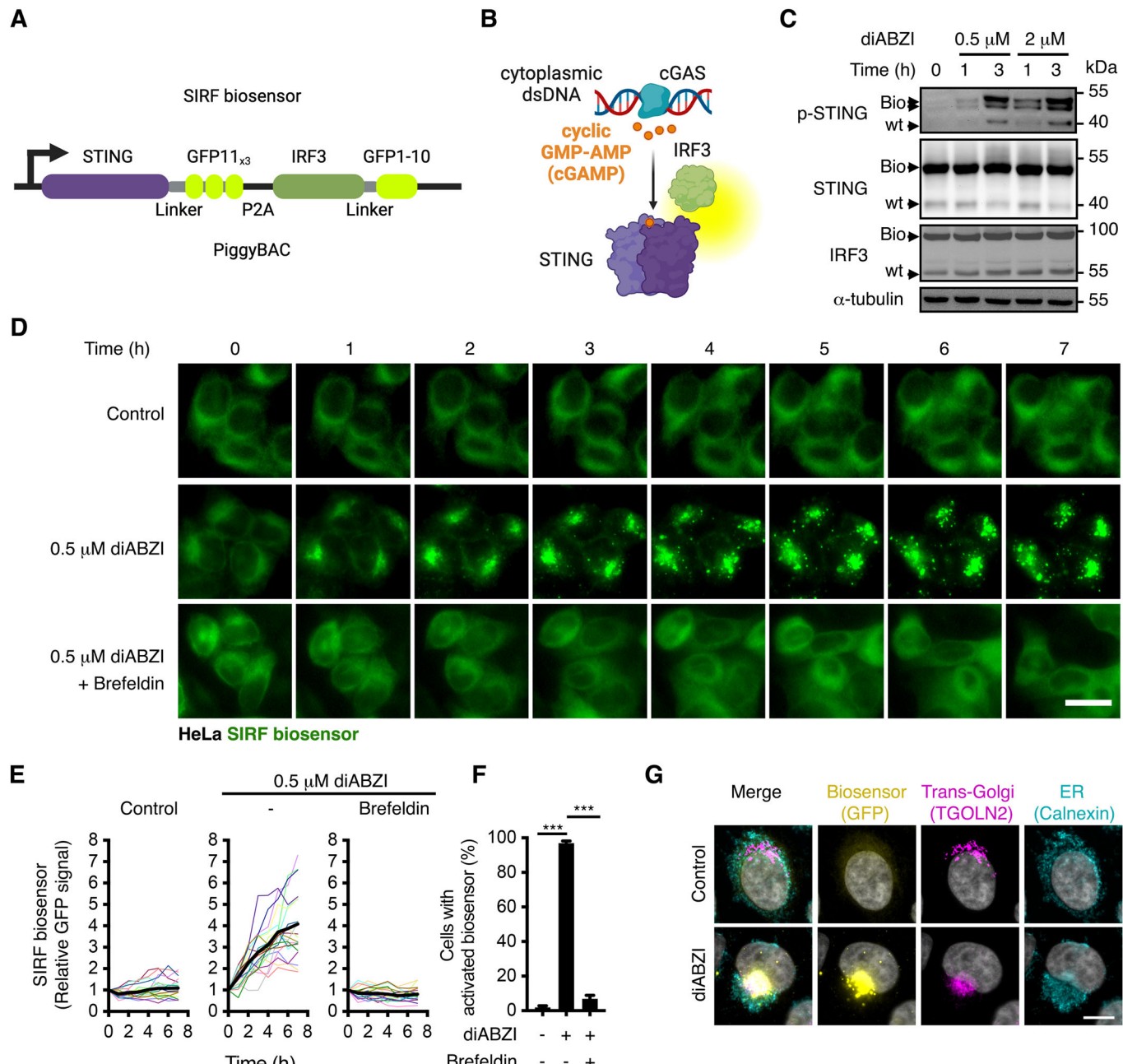

**Figure 1. Design and validation of a STING-IRF3 biosensor.**

(A, B) Schematics of the PiggyBAC construct containing the SIRF biosensor (A) and the suggested mode of function (B). (C) Western blot analyses in HeLa cells stably transfected with the SIRF biosensor upon treatment with the STING agonist diABZI. Bio, Biosensor construct; wt, wild type protein. Representative examples of $N = 3$ independent experiments are shown. (D–F) Live cell confocal imaging analyses of HeLa SIRF biosensor cells in the presence or absence of diABZI and the ER to Golgi transport inhibitor Brefeldin A. Note that each coloured line is one single cell randomly selected from the population and the black line is the mean of the analysed cells. *P*-values from one-way ANOVA between independent experiments from the indicated groups are indicated as ***$P < 0.0001$. In (F), activation of the biosensor was scored as clustering of the GFP signal in 1 or more puncta, and it is represented as mean ± SD of $N = 3$ independent time-lapses with >200 cells per condition. (G) Immunofluorescence co-localization analyses of HeLa SIRF biosensor cells in the presence or absence of diABZI for 4 h. Trans-Golgi: Anti-TGOLN2, ER: Anti-calnexin. Representative examples of $N = 3$ independent experiments are shown. Scale bar = 10 μm. Source data are available online for this figure.

Trans-Golgi (Fig. 1G, TGOLN2, consistent with a reported role of endolysosomal trafficking in STING signalling termination (Kuchitsu et al, 2023; Liu et al, 2022).

Live cell imaging analyses of HEK293T cells and fibroblasts (HFF-1) stably expressing the biosensor and treated with diABZI

were largely consistent with the studies in HeLa cells (Fig. EV1F,G and Movie EV3). Of note, HEK293T cells do not express cGAS or STING, allowing to uncouple the STING-IRF3 signalling activation from the dsDNA response (Fig. EV1G,H). HEK293T cells expressing the biosensor were able to transduce signalling and lead to

6-fold activation of the reporter upon 2 μM diABZI, similarly to HEK293T cells overexpressing wt STING (Fig. EV1G).

Titration experiments in HeLa cells with diABZI showed a similar time- and concentration-dependent response of the SIRF biosensor comparing with phospho-STING analyses by Western blots, but instead providing live information at single-cell resolution (Figs. 2A–C and EV2A–C). In particular, live cell imaging analyses showed that the biosensor clustered in HeLa cells treated with as little as 50 nM diABZI (Fig. 2A,B) and (ii) 78% of HeLa cells showed activation of the biosensor after just 45 min exposure to 2 μM diABZI (Fig. EV2A,B).

To avoid possible biases and time-consuming manual analyses, we set up two strategies for automated studies of the SIRF biosensor. First, using the free and open-source software CellProfiler (https://cellprofiler.org), we designed a pipeline for single-cell quantification of clustered biosensor signal (Integrated intensity/cell) in cells co-expressing H2B-mCherry, and imaged under the confocal microscope (Fig. 2D). CellProfiler analyses upon diABZI titration showed i) similar signal ratios across doses as in the manual quantification (Fig. 2A vs Fig. 2E), and ii) a slightly lower capacity of capturing active cells at reduced diABZI concentrations (Cells with activated biosensor at 0.05 and 0,1 μM diABZI, Fig. 2B vs Fig. 2E), due to a more stringent cut-off necessary for these automatic analyses (See methods). In summary, CellProfiler analyses of SIRF biosensor imaging allow for a fast and robust characterisation of large live cell imaging datasets. Second, we also performed automatic analysis of HeLa biosensor cells by live cell incubator imaging (IncuCyte), which allow for long-term culturing and monitoring of cells, as well as throughput analyses. IncuCyte automatic analyses reported dose- and time-dependent effects of diABZI, but underperformed Cellprofiler quantification of the biosensor under low diABZI concentration (0.05 and 0.1 μM diABZI; Fig. 2E,F), likely due to a higher background floor in this system. To further determine the suitability of the biosensor for high-throughput analyses, we performed siRNA assays in 384-well format using siRNA against TBK1 as positive control and imaging the plates in a large content microscope followed by CellProfiler analysis. Consistent with its requirement for STING activation, knock down of TBK1 consistently blocked the activation of the biosensor upon treatment with diABZI across all samples (Fig. EV2D–F).

Next, we examined the response of the biosensor directly to microenvironmental cGAMP, without cell permeabilization, thereby relying on its import by solute carriers (Cordova et al, 2021; Ritchie et al, 2019). Unlike the cell permeable STING agonist diABZI, microenvironmental cGAMP displayed a delay of 3–4 h in SIRF activation, possibly due to active import mechanisms, but reached similar dynamics and activation potential (Fig. 2G,H). Importantly, mutation to Alanines in the cGAMP binding residues R238/Y240 of STING and the dimerization-associated residue S386 of IRF3 fully abrogated the capacity of the SIRF biosensor to respond to microenvironmental cGAMP in HeLa cells (Fig. EV2G,H). Removal of cGAMP from the media led to a slow recovery of basal-like biosensor signal in HeLa cells within 24 h, similar to IRF3 nuclear-to-cytoplasm dynamics (Fig. EV2I,J). Of note, phospho-STING inactivation dynamics by Western blots were much faster even without agonist withdrawal (Fig. EV2K), as previously reported (Liu et al, 2022), indicating that the biosensor reports cellular signalling (active IRF3, Fig. EV2J), but cannot

accurately monitor the termination of receptor activation, possibly due to a longer life of the split GFP complex. Biosensor-activated cells displayed normal proliferation and viability, and could be normally passaged after withdrawal.

Taken together, these results show that the SIRF biosensor monitors the STING signalling response, including receptor activation and transport from ER, as well as TBK1 and IRF3 recruitment, and provides a fast (45 min), reversible, and single-cell resolution readout to STING agonists and microenvironmental cGAMP.

## Monitoring foreign dsDNA and intercellular cGAMP dynamics with a novel biosensor

To study how cGAMP spreads though a monolayer of cells, we first transfected HEK293T cells with either mCherry (Control) or cGAS-mCherry. Given that HEK293T cells do not harbour endogenous cGAS (Fig. EV1H), mCherry transfected cells should not produce cGAMP, while cGAS-mCherry transfected cells allow for self-triggering of the expressed cGAS by the transfected plasmid. Incubation cGAS-mCherry HEK293T cells with HEK293T SIRF biosensor cells resulted in a wave of activation of the biosensor starting from the cGAS-expressing cells that spread outwards 10–20 μm/h, while control mCherry cells induced no activation (Fig. 3A,B and Movie EV5). These results could be consistent with cGAMP being transfer through e.g., gap-junctions (Ablasser et al, 2013b; Chen et al, 2016) and/or vesicles instead of diluted in the media. Interestingly, cGAS-mCherry HEK293T cells also activated HeLa SIRF biosensor cells in a distance-dependent manner, albeit reaching a lower radius and speed (<5 μm/h) (Fig. 3C,D and Movie EV6). These results may account from differences in cell junctions and other paracrine/transfer methods, activation dynamics, as well as cell size and spacing between HEK293T and HeLa cells, and highlight the relevance of the biosensor to monitor cGAMP spreading across different cell populations.

Next, we examine the response of the biosensor to foreign sources of dsDNA. Transfection of plasmid DNA fluorescently labelled with Thiazole Red Homodimer (TOTO3) induced the activation of the SIRF biosensor in a concentration-dependent manner (Fig. 3E,F), and with similar dynamics as microenvironmental cGAMP (Fig. 2H). Comparison to qRT-PCR of IRF3 target genes showed that the biosensor response to dsDNA was as sensitive as *CXCL10* expression, and higher than interferon or *IL6* expression (Fig. EV3A). Of note, plasmid DNA transfected cells (TOTO3 positive) displayed a largely synchronised response starting 3 h post-transfection, followed by biosensor activation in untransfected neighbouring cells (TOTO3 negative) with ~2-h median delay compared to plasmid DNA carrying cells (Fig. 3E,G). The latter displayed similar dynamics as bystander HeLa cells co-cultured with cGAS-mCherry HEK293T cells (Fig. 3C,D).

To test the response of the biosensor to viruses, we exposed HeLa cells to 0.5 multiplicity of infection (MOI) of (i) the DNA virus HSV-1 expressing mCherry or (ii) the RNA virus influenza A expressing dsRed, and monitored the cells for 24 h by live cell incubator imaging. Infection with HSV-1, but not influenza A, led to a quick activation of the biosensor in HeLa cells (Figs. 4A,B and EV3B), showcasing the specificity of the biosensor for dsDNA. Further analyses showed that HeLa cells infected with HSV-1 (mCherry positive) displayed a heterogenous and largely low activation (10% of cells after 24 h) of their biosensors (Fig. 4C).

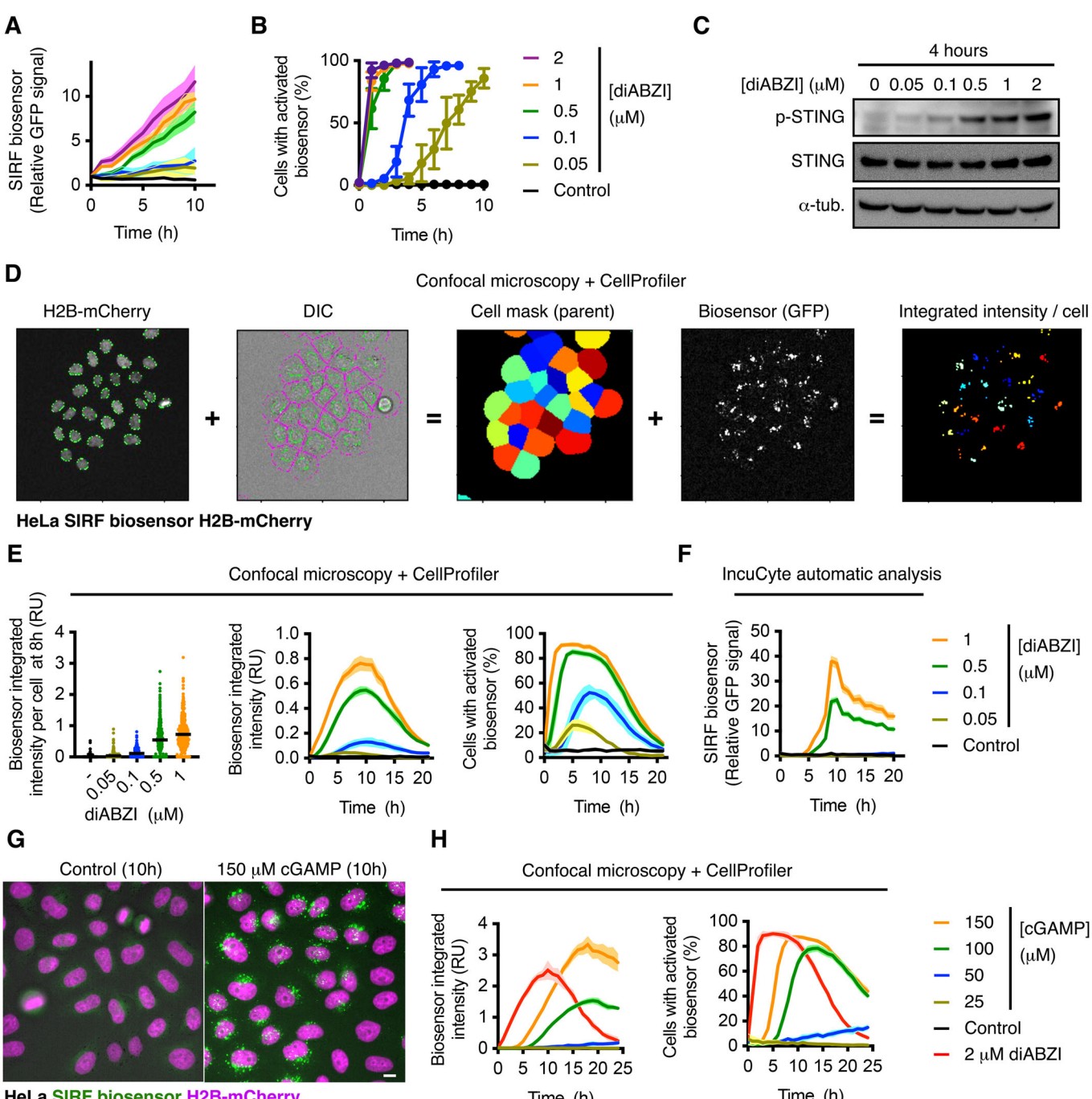

**Figure 2. Characterisation of the SIRF biosensor dynamics.**

(A, B) Live cell confocal imaging analyses of HeLa SIRF biosensor cells in the presence of the indicated concentrations of diABZI. In (A), each condition is showed as the mean ± SEM of 20 randomly selected cells. In (B), activation of the biosensor was scored as clustering of the GFP signal in 1 or more puncta >200 cells per condition, and is showed as mean ± SEM of active biosensor cells of $N = 4$ independent time-lapses. (C) Western blot analyses in wt HeLa cells upon treatment with the STING agonist diABZI. (D) CellProfiler pipeline for the automatic characterisation of the biosensor in SIRF biosensor H2B-mCherry HEK293T or HeLa cells. (E) CellProfiler analyses of HeLa SIRF biosensor H2B-mCherry cells after live cell imaging in the presence of the indicated concentrations of diABZI. Data represent mean ± SEM of 4 different parallel time-lapses per condition each automatically analysed and with 270–500 cells. (F) IncuCyte cell population imaging analyses of HeLa SIRF biosensor cells treated as in (E). Data represent mean ± SEM of relative fluorescence signal of 6 different parallel time-lapses with >200 cells per imaged field, and performed following manufacturer instructions (See methods). (G, H) Live cell confocal imaging of HeLa SIRF biosensor H2B-mCherry cells in the presence or absence of different concentrations of cGAMP. In (H), data was automatically analysed with CellProfiler and represent mean ± SEM of 6 different parallel time-lapses per condition each and with >300 cells. Representative examples of at least $N = 3$ independent experiments are shown, except for (F) which represents $N = 2$. Scale bar = 10 µm. Source data are available online for this figure.

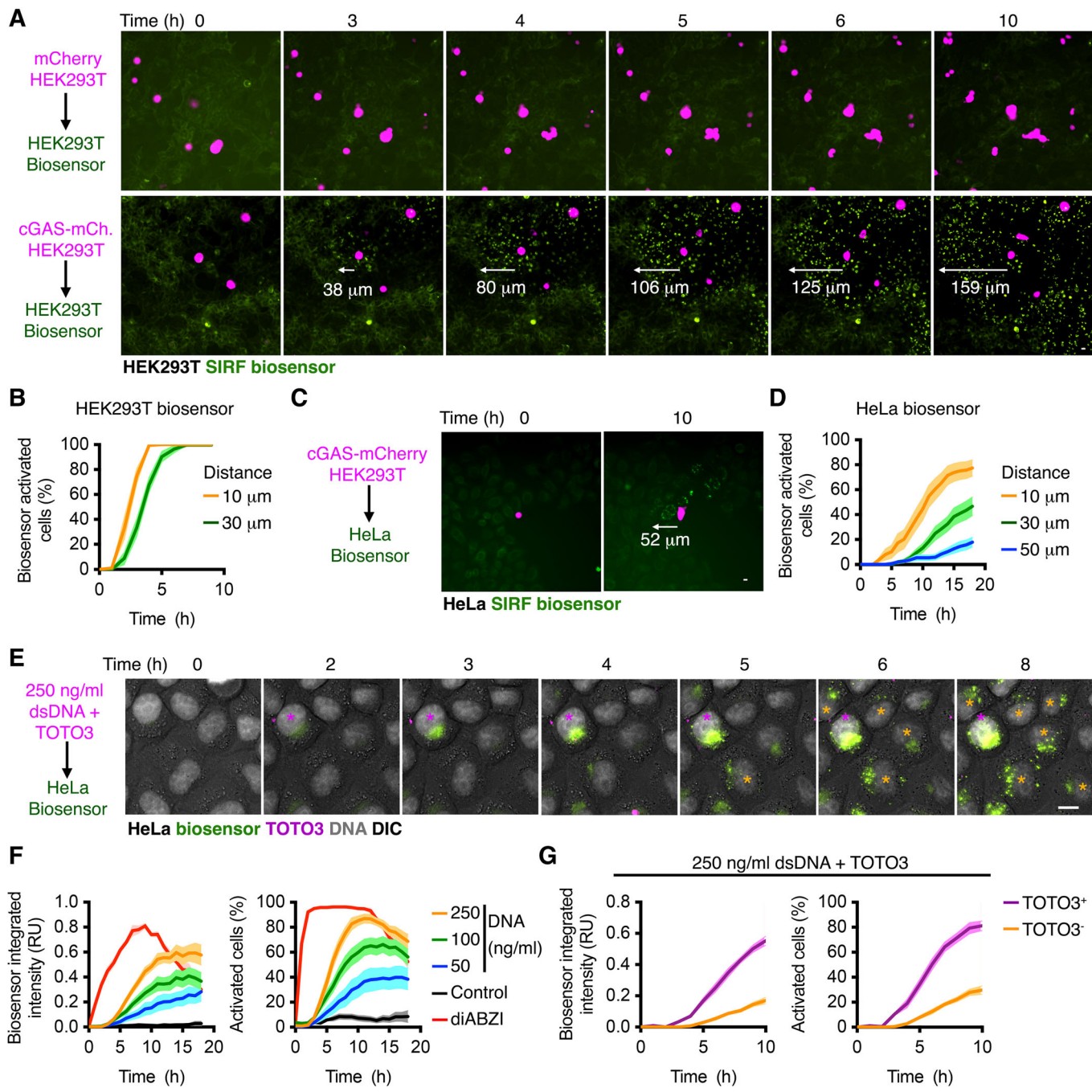

**Figure 3. Monitoring cGAMP transfer and foreign dsDNA with the SIRF biosensor.**

(A–D) Live cell confocal imaging analyses of SIRF biosensor HEK293T (A, B) and HeLa (C, D) cells co-cultured with mCherry (Control) or mCherry-cGAS HEK293T cells. In (B, D), data represents mean ± SEM of the percentage of activated SIRF biosensor cells surrounding $N = 19$–20 mCherry[+] cells within the indicated distances (10 µm = 0–10 µm; 30 µm = 10–30 µm; 50 µm = 30–50 µm). (E–G) Live cell confocal imaging analyses of HeLa SIRF biosensor H2B-mCherry cells transfected with different concentrations of a circular plasmid fluorescently labelled with TOTO3 (note that the TOTO3 signal is saturated to ensure we show all transfected cells). In (E), a magenta asterisk marks the activation in a transfected cell, and orange asterisks mark activated bystander cells (TOTO3⁻). In (F), CellProfiler analyses of HeLa SIRF biosensor H2B-mCherry cells represent mean ± SEM of 6 different parallel time-lapses with >300 cells from per condition and automatically analysed. In (G), CellProfiler data from (F) was further categorised and analysed for TOTO3 positive and negative cells (See methods). Representative examples of $N = 3$ independent experiments are shown. Scale bars = 10 µm. Source data are available online for this figure.

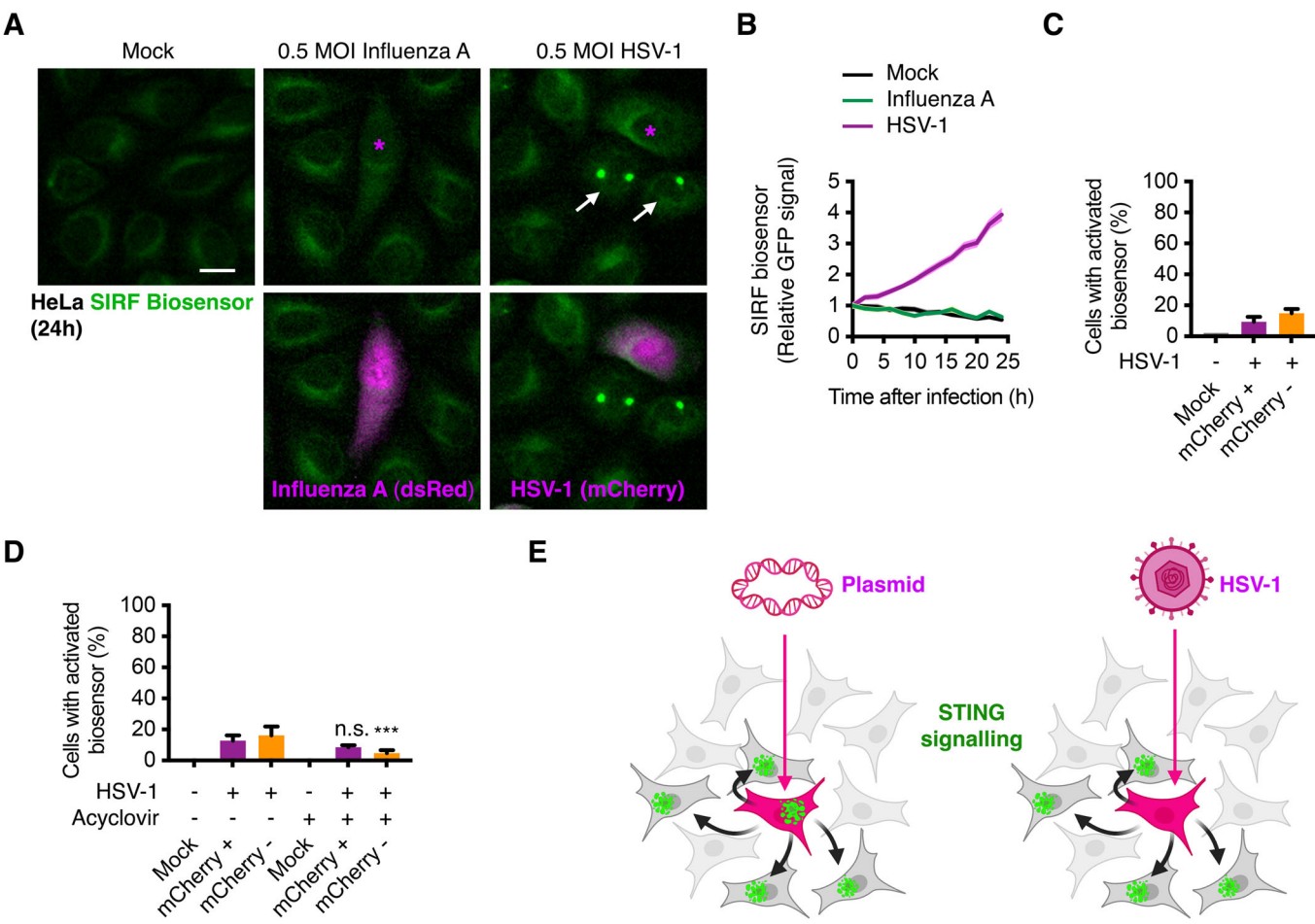

**Figure 4.  Characterisation of HSV-1 infection with the SIRF biosensor.**

(A–C) Live cell incubator imaging analyses of HeLa SIRF biosensor cells mock treated (Control) or infected with 0.5 MOI HSV-1 expressing mCherry or influenza A expressing dsRed. In (A), magenta asterisks mark HSV-1(mCherry⁺) or Influenza A (dsRed⁺) infected cells during the 24-h recording, and white arrows mark cells with clearly activated biosensor. In (B), population fluorescence data was automatically analysed using IncuCyte software. In (C), the activated biosensor was scored as clustering of the GFP signal in 1 or more puncta either infected (mCherry⁺) or bystander (mCherry⁻) cells and is showed as mean ± SD of N = 6 time-lapses with >200 cells per condition. (D) HeLa SIRF biosensor cells mock treated or infected with 0.5 MOI HSV-1 expressing mCherry after 24 h, in the presence or absence of acyclovir and analysed as described in (C). P-values from one-way ANOVA between biological replicates and compared to control are indicated as ***P = 0.0002 and n.s. = 0.076. (E) model of proposed intrinsic and extrinsic STING activation upon exogenous dsDNA. Scale bars = 10 μm. Source data are available online for this figure.

Intriguingly, 15% uninfected neighbours responded to the population infection (Fig. 4C), which is in line with the bystander cell activation upon dsDNA transfection (Fig. 3G). These results could support previous findings showing that viral infections can hamper the intrinsic cellular STING and other anti-viral responses, thereby leading to innate immune evasion (Deschamps and Kalamvoki, 2017; Drayman et al, 2019; Hare et al, 2020; Sun et al, 2015), and highlights that microenvironmental spread of cGAMP and/or released dsDNA to the neighbouring cells can be a source of STING activation in an infected cell population. Inhibition of viral replication by acyclovir reduced 3-fold the bystander response (Fig. 4D), suggesting that HSV-1 replication rather the infection itself is a major driver of the population response.

These results highlight the suitability of this novel tool to monitor the heterogenous innate immune response via cGAS-STING signalling towards foreign DNA, especially in the context of viral infections (Fig. 4E).

## Monitoring host cytoplasmic dsDNA with a novel SIRF biosensor

To determine the response of the biosensor to intrinsic sources of dsDNA, we performed apoptosis, TREX1 depletion, DNA replication stress and chromosome missegregation experiments.

We induced apoptosis in HeLa cells using a cocktail of BH3-mimetics (see methods) that induce BAX/BAK activation by specifically blocking their inhibitors and monitored the kinetics of cell death for 24 h by live cell incubator imaging using the internalization of DRAQ7 as a proxy. BAX/BAK activation induces the opening of the apoptotic pore and the permeabilization of the mitochondrial outer membrane, which is followed by release of mtDNA into the cytosol (Cosentino et al, 2022; McArthur et al, 2018; Riley et al, 2018). In agreement with previous studies showing that activated apoptotic caspases cleave and inactivate cGAS thus preventing the engagement of the innate immune response in

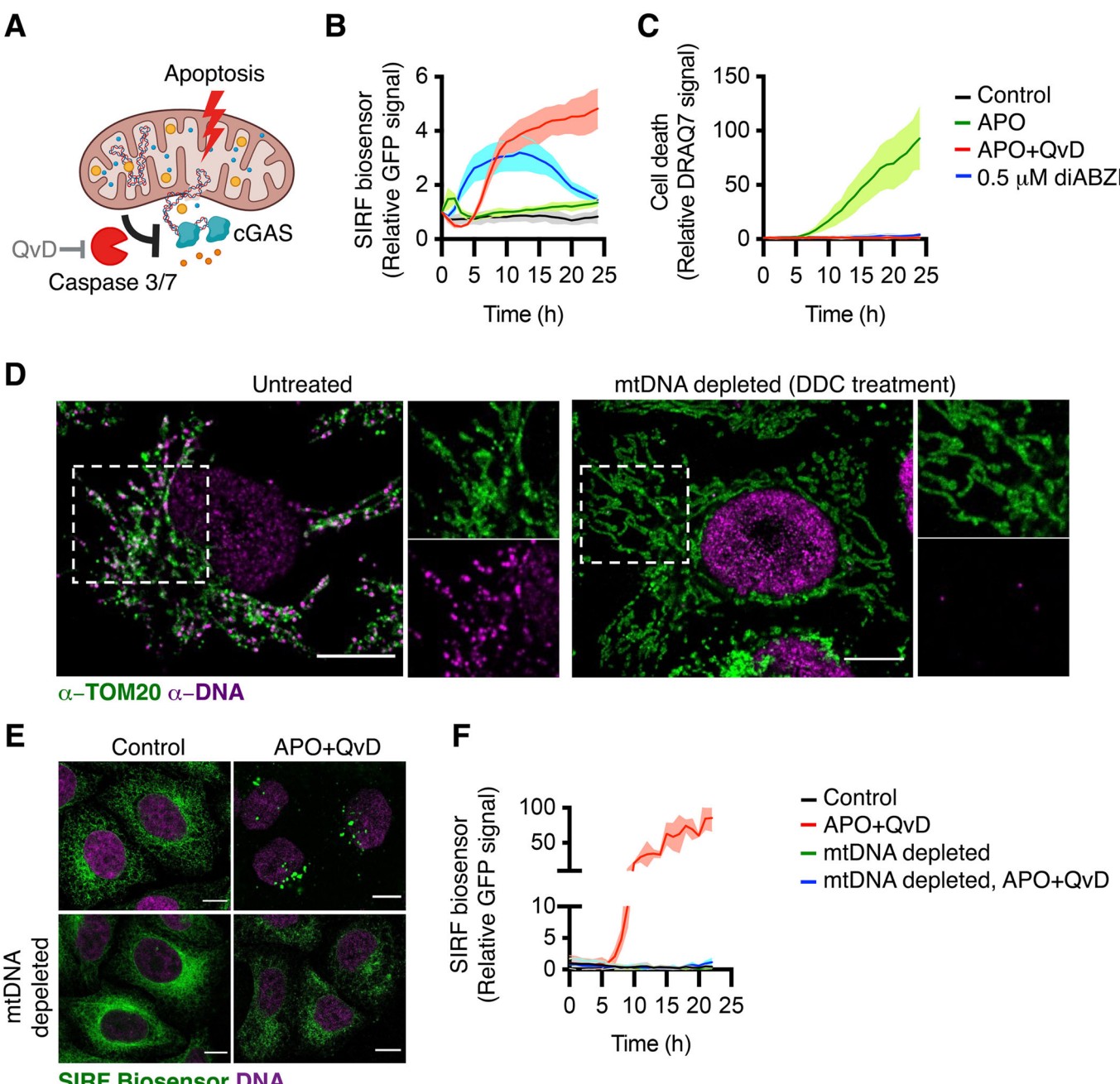

**Figure 5. Monitoring mtDNA flux upon apoptosis.**

(A) Model of the caspase-dependent inhibition of cGAS during apoptosis. (B, C) Cell population incubator imaging analyses of HeLa SIRF biosensor cells in the presence or absence of an apoptotic cocktail of BH-3 mimetics, diABZI, with or without the caspase inhibitor QvD. In (B, C), population fluorescence data was automatically analysed using IncuCyte software. In (C), cell death was quantified by cell permeabilization towards DRAQ7. (D, E) Immunofluorescence experiments in HeLa SIRF biosensor cells after 1-week treatment with DDC, or mock treated (Control). (F) IncuCyte cell population imaging analyses of HeLa SIRF biosensor cells treated with DDC for 1 week (mtDNA depleted) or mock treated (Control) in the presence or absence of an apoptotic cocktail and the caspase inhibitor QvD. Scale bars = 10 μm. Source data are available online for this figure.

response to cytosolic mtDNA (Cosentino et al, 2022; Giampazolias et al, 2017; White et al, 2014), triggering of apoptosis in HeLa cells resulted in cell death, but not activation of the biosensor (Fig. 5A–C). Importantly, inhibition of caspase activity with the pan-caspase inhibitor QVD-OPH (QvD) not only prevented cell death (Fig. 5C), but also led to the activation of the biosensor, which was detectable 5 h

after treatment in >90% of the treated HeLa cells (Fig. 5B and Movie EV7). To confirm that the biosensor indeed responded to released mtDNA during apoptosis and not to other activators, we treated HeLa cells with 2′,3′-dideoxycytidine (DDC) for one week, which removes mtDNA by preventing its replication (Fig. 5D) (Chen and Cheng, 1989). In contrast to untreated cells, DDC-treated cells did

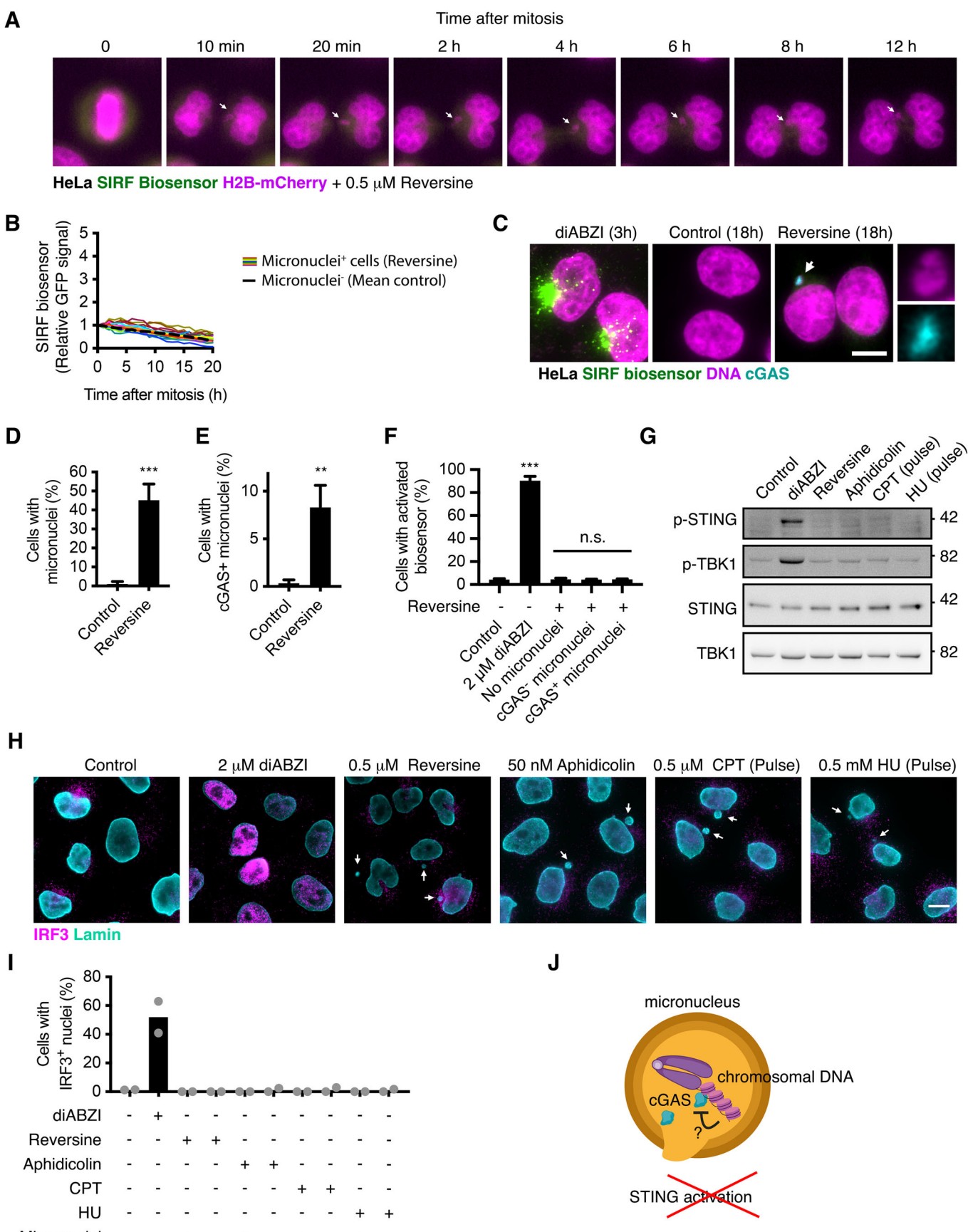

**A** Time after mitosis

0  10 min  20 min  2 h  4 h  6 h  8 h  12 h

**HeLa SIRF Biosensor H2B-mCherry + 0.5 μM Reversine**

**B**

SIRF biosensor (Relative GFP signal) vs Time after mitosis (h)

▬ Micronuclei⁺ cells (Reversine)
▬ Micronuclei⁻ (Mean control)

**C** diABZI (3h)  Control (18h)  Reversine (18h)

**HeLa SIRF biosensor DNA cGAS**

**D** Cells with micronuclei (%)

Control  Reversine  ***

**E** Cells with cGAS+ micronuclei (%)

Control  Reversine  **

**F** Cells with activated biosensor (%)

Reversine  −  −  +  +  +

Control  2 μM diABZI  No micronuclei  cGAS⁻ micronuclei  cGAS⁺ micronuclei

*** n.s.

**G**

Control, diABZI, Reversine, Aphidicolin, CPT (pulse), HU (pulse)

p-STING — 42
p-TBK1 — 82
STING — 42
TBK1 — 82

**H**

Control  2 μM diABZI  0.5 μM Reversine  50 nM Aphidicolin  0.5 μM CPT (Pulse)  0.5 mM HU (Pulse)

**IRF3 Lamin**

**I** Cells with IRF3⁺ nuclei (%)

| | | | | | | | | | | |
|---|---|---|---|---|---|---|---|---|---|---|
| diABZI | − | + | − | − | − | − | − | − | − | − |
| Reversine | − | − | + | + | − | − | − | − | − | − |
| Aphidicolin | − | − | − | − | + | + | − | − | − | − |
| CPT | − | − | − | − | − | − | + | + | − | − |
| HU | − | − | − | − | − | − | − | − | + | + |
| Micronuclei | − | − | + | − | + | − | + | − | + | − |

**J** micronucleus

chromosomal DNA

cGAS  ?

STING activation

**Figure 6. Micronuclei do not activate the STING signalling.**

(A, B) Live cell confocal imaging analyses of HeLa SIRF biosensor H2B-mCherry cells treated with 0.5 µM reversine, undergoing chromosome missegregation and micronuclei formation during the first 3 h of the recording, and monitored for 17–20 h. In (A), white arrows mark a lagging chromosome and the resulting micronucleus. (C–F) Immunofluorescence experiments in HeLa SIRF biosensor cells treated with 0.5 µM reversine for 18 h. In (C), the arrow indicate a micronucleus. In (F), the activated biosensor was scored as clustering of the GFP signal in 1 or more puncta in cells treated as indicated (diAZI, reversine) and harbouring micronuclei without or with cGAS (ruptured). Data is showed as mean ± SD of $N = 3$ independent experiments with >200 cells per condition. In (D, E), $P$-values from a t-test between the two groups with $N = 3$ independent experiments are indicated as ***$P = 0.0009$ and **$P = 0.0041$, respectively. In (F), $P$-values from one-way ANOVA between the indicated groups with $N = 3$ independent experiments are indicated as ***$P < 0.0001$, or not significant (n.s.) from left to right as $P = 0.9998$, $P = 0.9999$, $P = 0.9999$. (G–I) Western blot (G) and immunofluorescence (H, I) analyses of HeLa cells treated with 2 µM diABZI for 4 h, with 0.5 µM reversine or 50 nM aphidicolin for 18 h, or pulsed for 4 h with 0.5 µM CPT or 0.5 mM HU followed by 18 h release. In (H), arrows indicate micronuclei. In (I) nuclear IRF3 was quantified both in cells with or without micronuclei, and data is mean of $N = 2$ independent experiments with >200 cells per condition. (J) Ruptured micronuclei do not activate STING signalling, possibly due to inhibitory roles of the nucleosomes. $P$-values from one-way ANOVA between $N = 3$ independent experiments from the indicated groups are indicated as *$P < 0.05$, **$P < 0.01$, ***$P < 0.001$, or n.s., not significant. Scale bars = 10 µm. Source data are available online for this figure.

not activate the biosensor during apoptosis, including in the presence of the caspase inhibitors (Fig. 5E,F). These results indicate that the efflux of mtDNA leads to cGAS-dependent biosensor activation, and support its reliability to explore e.g., the dynamics and signalling consequences of mitochondria outer membrane pores during apoptosis.

Chromosome segregation errors such as lagging chromosomes can lead to formation of micronuclei (Krupina et al, 2021). Rupture of the micronuclei's nuclear envelope has shown to trigger cGAS recruitment (Harding et al, 2017; Mackenzie et al, 2017; Martin et al, 2024), and it has been proposed as a critical mechanism of surveillance in the context of chromosomally instable (CIN) tumours (Hong et al, 2022). Other reports point towards chromatin bridges triggering cGAS activation instead (Flynn et al, 2021). To examine this question, we treated SIRF biosensor mCherry-H2B HeLa cells with the MPS1 inhibitor reversine (Santaguida et al, 2010), which blocks the spindle assembly checkpoint thereby resulting in whole-chromosome missegregation and micronuclei (Martin et al, 2024; Santaguida et al, 2010).

We treated HeLa cGAMP-biosensor H2B-mCherry cells with 0.5 µM reversine, and monitored (i) chromosome missegregation and micronuclei formation during the first 3 h, and (ii) biosensor activity during 24 h by live cell imaging. Unexpectedly, HeLa cells harbouring micronuclei after mitosis did not show activation of the biosensor during the following 17–20 h of the recording (Fig. 6A,B). Given that a large proportion of micronuclei might not rupture during the recording, we performed immunofluorescence analyses in HeLa biosensor cells after 18 h treatment with reversine, which resulted in ~33% of cells with cGAS⁻ micronuclei (likely intact) and ~7% of cells with cGAS⁺ micronuclei (ruptured) (Fig. 6C–E). Consistent with their whole-chromosome segregation origin, 92% of micronuclei were positive for the centromere protein CENPC (Fig. EV4A,B). Intriguingly, neither HeLa cells with cGAS⁺ micronuclei, nor their neighbours, displayed activation of the SIRF biosensor compared to control cells treated with diABZI (Fig. 6C,F and Movies EV8, 9), and other sources of cytoplasmic dsDNA (Figs. 3–5 and EV4). In addition to reversine, we also generated micronuclei by inducing mild DNA replication stress and subsequent chromosome missegregation and ultra-fine bridges using 50 nM aphidicolin (Bohly et al, 2022; de Jaime-Soguero et al, 2024), as well as upon a pulse of 0.5 µM camptothecin (CPT) or 0.5 mM hydroxyurea (HU) (Fig. EV4C,D). Neither treatment induced STING phosphorylation in Western blot analyses (Fig. 6G),

despite consistently generating micronuclei (Fig. EV4C,D). In agreement, none of the treatments induced IRF3 nuclear translocation either in micronuclei positive cells or their neighbours (micronuclei negative) (Fig. 6H,I). Similar as in the case of reversine, live cell imaging analyses of HeLa SIRF biosensor H2B-mCherry cells pulsed with 0.5 µM CPT did not reveal activation of the biosensor in micronuclei positive cells (Fig. EV4E). However, we detected activation of the biosensor after 10–15 h pulse with CPT in dying cells with signs of autophagy, independently of whether they harboured micronuclei (Fig. EV4F), possibly due to STING roles autophagy (Kuchitsu et al, 2023; Xun et al, 2024), and explaining previous reports linking topoisomerase poisoning (Marinello et al, 2022), as well as other sources of DNA replication stress and damage, to STING signalling activation. Indeed, induction of high replicative stress with continuous treatment of 0.5 µM Aphidicolin (i) led to similar activation of the biosensor in dying cells compared to a CPT pulse (Fig. EV4F,G), and (ii) also failed to activate STING or IRF3 in parallel Western blot and immunofluorescence analyses (Fig. EV4H,I). Intriguingly, 0.5 µM Aphidicolin induced TBK1 phosphorylation and expression of *CXCL10* within 24 h (Fig. EV4H,J), suggesting that other pattern recognition pathways may underlie the innate immune response to DNA damage and/or micronuclei. Accordingly, reversine treatment induced *CXCL10* expression not only in HeLa cells, but also in HEK293T cells (Fig. EV4K), which lack STING and cGAS (Fig. EV2G,H).

Finally, misregulation of DNA-degrading enzyme TREX1 is associated with aberrant inflammation and auto-immunity (Li et al, 2024; Simpson et al, 2020). The precise origins of dsDNA targeted by TREX1 is not fully understood, and could account for either gDNA released from the nucleus or dsDNA taken up from the microenvironment (Simpson et al, 2020). Knock down of TREX1 led to a robust activation of the biosensor in 16% of HeLa cells (Fig. EV5A,B), which was accompanied with abnormal nuclear morphology (Fig. EV5C), possibly due to roles of TREX1 in nuclear envelope maintenance (Nader et al, 2021). These results highlight the relevance of the biosensor to monitor the consequences of pathological TREX1 misregulation in the innate immune response. Previous research highlighted a role of TREX1 in cGAS inhibition at micronuclei by competing against cGAS to degrade DNA (Mohr et al, 2021). However, knock down of TREX1 did not significantly impact the activation of the biosensor in post-mitotic HeLa cells harbouring ruptured micronuclei (Fig. EV5E,F).

Taken together, these results indicate that the SIRF biosensor is suitable to monitor host cytoplasmic dsDNA such as mtDNA flux during apoptosis or TREX1 deficiency, and suggest that ruptured micronuclei do not per se activate STING signalling, possibly due to cGAS sequestration and inactivation by the nucleosome histones H2A/B (Michalski et al, 2020; Mohr et al, 2021; Pathare et al, 2020) (Fig. 6J).

## Discussion

Here, we presented a fluorescent biosensor (SIRF) that can robustly react within 45 min and in a time- and concentration-dependent manner to STING agonists and cGAMP. We demonstrate that the SIRF biosensor is suitable for single-cell characterisation of the innate immune signalling dynamics upon foreign dsDNA, allowing to study mechanisms of innate immune evasion by viruses (Deschamps and Kalamvoki, 2017; Sun et al, 2015), as well as their recognition, including signalling spread through a cell population. Among other sources of cytoplasmic dsDNA, we also show that the SIRF biosensor serves to monitor mtDNA release and signalling blockage by caspases following apoptosis. We also demonstrated that it can report TREX1 deficiency (Fig. EV5), which associated with several autoimmune diseases (Li et al, 2024; Simpson et al, 2020; Stetson et al, 2008). Given its robustness, this biosensor could help in further studying the contribution of mtDNA to inflammatory signalling associated with disease under different conditions, like mitochondrial dysfunction, cancer, and neurodegeneration (Riley and Tait, 2020).

Unlike mtDNA released to the cytoplasm, we show that cGAS recruited to ruptured micronuclei failed to activate the SIRF biosensor. Each HeLa cell contains ~140 Mbp of total mtDNA (Bogenhagen and Clayton, 1974), while the average chromosome contains ~130 Mbp. Thus, given the robust response of the biosensor to apoptosis in the absence of active caspases (Fig. 4), the lack of response upon chromosome missegregation cannot be attributed to low levels of dsDNA in the micronuclei, which largely contain full chromosomes in the case of reversine treatments (Fig. EV4A), but possibly to how the DNA presented. In that respect, our results are consistent with recent structural reports showing that the nucleosome histones H2A/B inhibit cGAS activation towards genomic DNA (Michalski et al, 2020; Pathare et al, 2020).

In follow up analyses, we found that the induction of micronuclei upon DNA replication stress or inhibition of the spindle assembly checkpoint neither induced phosphorylation of STING nor IRF3 translocation into the nuclei, which are in contrast with previous reports linking micronuclear cGAS to STING-dependent activation of interferon signalling (Harding et al, 2017; Mackenzie et al, 2017; Mohr et al, 2021). These discrepancies could be attributed to (i) focus on cGAS recruitment, instead of STING activation; (ii) secondary activation of cGAS-STING due to cell death (See Fig. EV4F), chromatin bridges and/or mitotic arrest (Flynn et al, 2021; Zierhut et al, 2019); (iii) activation of the interferon response upon chronic exposition or through other molecular cascades. In that respect, we show that micronuclei formation induced the expression of the cytokine *CXCL10* even in cells lacking STING and cGAS (Fig. EV4K,L). Taken together, our results indicate that ruptured micronuclei generated by whole-chromosome missegregation or under-replicated DNA (e.g., ultra-fine bridges) are poor activators of STING signalling, possibly due to the inhibitory role of the nucleosomes, and suggest that other cytosolic pattern recognition receptors may underlie the interferon response upon chromosomal instability (Note: During the preparation of this manuscript two studies reported that micronuclei and—in general—nucleosome packed DNA can induce interferon response but do not activate STING signalling (Sato and Hayashi, 2024; Takaki et al, 2024), and supported a previous study that focused instead on chromatin bridges (Flynn et al, 2021)).

Furthermore, we showed that the biosensor sensitivity is sufficient to report microenvironmental cGAMP, to monitor its intercellular wave-like transfer through bystander cells over long distance (>150 μm), as well as to unveil the spread of this immune-transmitter across a cell population following viral infection, with relevance for future studies on how the innate immune response unrolls across tissues.

Taken together, we provide a toolset to (i) generate reporter cell lines in a single step; (ii) easily monitor the activation cGAS-STING-IRF3 signalling pathway by live cell imaging, incubator imaging, or immunofluorescence; (iii) automated analyses towards high content imaging; and (iv) capture the spatio-temporal and heterogenous dynamics of the response to cGAMP at single-cell resolution.

## Methods

**Reagents and tools table**

| Reagent/resource | Reference or source | Identifier or catalog number |
|---|---|---|
| **Experimental models** | | |
| HEK293T cells | ATCC | CRL-3216 |
| HeLa cells | ATCC | CCL-2 |
| HFF1 cells | ATCC | SCRC-1041 |
| HeLa SIRF biosensor | This study | |
| HeLa SIRF biosensor H2B-mCherry | This study | |
| HeLa SIRF biosensor (mutant) | This study | |
| HEK293T SIRF biosensor | This study | |
| HSV-1 expressing mCherry HSV1(17+)Lox-CheP2AGLuc | (Devadas et al, 2014) | B. Sodeik |
| Influenza A SC35M_NS1_2A_dsRED | – | M. Schwemmle |
| **Recombinant DNA** | | |
| PiggyBac(Puro) | BioCat | PB510B-1-SBI |
| PiggyBac Transposase | BioCat | PB210PA-1-SBI |
| SIRF Biosensor | This study | Methods section |
| SIRF Biosensor (mutant) | This study | Methods section |
| SV40NLS-mCherry | This study | Methods section |
| cGAS-mCherry | This study | Methods section |

| Reagent/resource | Reference or source | Identifier or catalog number |
|---|---|---|
| IFN-Beta_pGL3 | Addgene | 102597 |
| pCS2+ | Berger et al, 2017 | Christof Niehrs |
| WT STING | This study | Methods section |
| **Antibodies** | | |
| Mouse anti α-tubulin | Sigma-Aldrich | T9026 |
| Rabbit anti-TBK1 | Cell Signaling Technologies | 38006 |
| Rabbit anti-STING | Cell Signaling Technologies | 13647 |
| Rabbit anti pTBK1 | Cell Signaling Technologies | 5483 |
| Rabbit anti pSTING | Cell Signaling Technologies | 50907 |
| Rabbit anti-IRF3 | Cell Signaling Technologies | 11904 |
| Rabbit anti-cGAS | Cell Signaling Technologies | 15102 |
| Rabbit anti-cGAS | Cell Signaling Technologies | 79979 |
| Rabbit anti-TREX1 | Cell Signaling Technologies | 15107 |
| Goat anti-mouse IgG HRP | Millipore | AP308P |
| Goat anti-rabbit IgG HRP | Cell Signaling Technologies | 7074 |
| Rabbit anti-TGOLN2 | Cell Signaling Technologies | 55727 |
| Mouse anti-Calnexin | Santa Cruz | sc23954 |
| Mouse anti LaminA/C | Cell Signaling Technologies | 4777 |
| Guinea pig anti-CENP-C | MBL International | PD030 |
| Donkey anti-guinea pig Cy3 | Millipore | AP193C |
| Donkey anti-rabbit Alexa594 | ThermoFischer | A21206 |
| Donkey anti-rabbit Alexa647 | ThermoFischer | A31573 |
| Mouse anti-DNA | Progen | 690014S |
| Rabbit anti-TOMM20 | Atlas | HPA011562 |
| STAR ORANGE | Abberior | STORANGE-1001 |
| STAR RED | Abberior | STRED-1002 |
| **Oligonucleotides and other sequence-based reagents** | | |
| qPCR Primers | – | Methods section |
| Cloning Primers | – | Methods section |
| siScramble | Sigma-Aldrich | SIC001 |
| siControl | Dharmacon | D-001206-14-20 |
| siTBK1 | Dharmacon | M-003788-02-0010 |
| siTREX | Dharmacon | M-013239-03-0005 |

| Reagent/resource | Reference or source | Identifier or catalog number |
|---|---|---|
| **Chemicals, Enzymes and other reagents** | | |
| DAPI | Sigma-Aldrich | 10236276001 |
| S63845 | Hölzel | HY-100741 |
| ABT-747 | Hölzel | HY-50907 |
| Q-VD-Oph | Hölzel | HY-12305g |
| TOTO-3 | ThermoFischer | T3604 |
| diABZI (Compound 3) | Invivogen | tlrl-diabzi-2 |
| 2'3' cGAMP | Invivogen | tlrl-nacga23-1 |
| Lipofectamine3000 | ThermoFischer | L3000001 |
| Lipofectamine RNAiMAX | ThermoFischer | 13778075 |
| Xt-remeGENE 9 | Sigma-Aldrich | 06 365 787 001 |
| Brefeldin A | (Berger et al, 2017) | – |
| Reversine | Sigma-Aldrich | R3904-1MG |
| Camptothecin | Selleckchem | NSC-100880 |
| Hydroxyurea | Sigma-Aldrich | H8627-1G |
| Aphidicolin | Santa Cruz | sc-201535A |
| Acyclovir | MedChemExpress | HY-17422 |
| 2',3' dideoxycytidine (DDC) | (Chen and Cheng, 1989; Nelson et al, 1997) | |
| **Software** | | |
| Cell Profiler | CellProfiler | |
| Fiji | ImageJ | |
| RStudio | Posit | |
| Bfconvert | GitHub | VolkerH |
| **Other** | | |
| Nikon-Ti2 | Nikon | |
| Inverted Nikon Ti2 | Nikon | |
| IncuCyte S3 | Sartorius | |
| IncuCyte SX5 | Sartorius | |

## Constructs

The SIRF biosensor utilises tandem split GFP (Kamiyama et al, 2016). In detail, a construct containing STING-GFP11$_{x3}$ (GFP(N)) and IRF3-GFP1-10 (GFP(C)) were cloned into the PiggyBAC backbone (Biocat PB510B-1-SBI). separated by a ribosome skipping peptide (P2A) to facilitate stochiometric expression (Referred from here on as SIRF biosensor).

The SIRF biosensor (mutant) was constructed by mutating the following sequences: STING R238A/Y240A and IRF3 S386A from the original SIRF biosensor sequence. The new insert was cloned back into the PiggyBAC backbone.

SV40NLS-mCherry was constructed by purchasing two oligos of the SV40NLS sequence with "cut" SacI and BamHi sequences at both ends: Fw: 5′ cgccaccatgcccaagaagaagaggaaagtcggg 3′ and Rev: 5′ gatccccgactttcctcttcttcttgggcatggtggcgagct 3′.

Both primers in nuclease-free water and obtaining a final volume of 50 µl and a final concentration of each primer at 5 µM.

Annealing was done by placing the mixed primers in a thermocycler at 95 °C for 5 min and then cooling down at steps of 0.12 C/s until 12 °C was reached. Annealed oligos were purified via column purification. "Cut" SV40NLS fragment was then cloned into a pCS2+ plasmid containing mCherry.

cGAS-mCherry was constructed by amplifying the wild-type cGAS sequence and cloning it into a pCS2+ plasmid containing mCherry.

WT STING plasmid was generated by linking the SFFV promoter to the wild-type STING sequence and was cloned into pTBL209 pcDNA Cas9-T2A-TdT (Addgene: 126424).

The construct and maps are available at request.

## Cell culture

HeLa and HEK293T cells (ATCC) were cultured in DMEM medium (Gibco) supplemented with 10% FBS and 1% penicillin/streptomycin.

To generate the HeLa and HEK293T SIRF biosensor cells, the SIRF biosensor plasmid was co-transfected with the PiggyBAC transposase plasmid (Biocat PB210PA-1-SBI) at a ratio of 500 ng to 100 ng, respectively, in a six-well plate. Transfection was carried out with Lipofectamine3000 (ThermoFisher) according to manufacturer's protocol. After 72 h cells were selected with 2 µg/ml and 3 µg/ml puromycin for the HeLa and HEK293T cell lines, respectively. Monoclonal cell lines were then expanded from the surviving cells.

To generate the HeLa SIRF biosensor H2B-mCherry cells, H2B-mCherry was cloned into the PiggyBAC backbone plasmid (Biocat PB510B-1-SBI). The H2B-mCherry plasmid was then co-transfected with the PiggyBAC transposase plasmid at a ratio of 500 ng to 100 ng, respectively, into HeLa SIRF biosensor cells in a six-well plate. Transfection was carried out with Lipofectamine3000 (ThermoFisher) according to manufacturer's protocol. After 72 h, the H2B-mCherry positive HeLa SIRF biosensor cells were pooled together via cell sorting and then monoclonal cell lines were generated from the polyclonal cells. Activation of the biosensor did not impact cell viability (Movie EV10) and cells could be passaged normally after withdrawal of e.g., diAZBI stimulation.

To generate the SIRF biosensor (mutant) cell line, HeLa cells were co-transfected with the SIRF biosensor (mutant) plasmid and PiggyBAC transposase plasmid at a ratio of 500 ng to 100 ng, respectively, in a six-well plate. Transfection was carried out with Lipofectamine3000 according to manufacturer's protocol. After 72 h, the cells were selected with 1 µg/ml puromycin. Polyclonal lines were used for the experiment.

To generate the SIRF biosensor HFF1 cells, the SIRF biosensor was co-transfected with the PiggyBAC transposase plasmid at a ratio of 500 ng to 500 ng, respectively, in a six-well plate. Transfection was carried out with Lipofectamine3000 according to manufacturer's protocol. After 72 h, cells were selected with 1 µg/ml puromycin and polyclonal cells were used for experiments.

Where indicated, mtDNA depletion was achieved via 2′,3′ dideoxycytidine (DDC) treatment (Chen and Cheng, 1989; Nelson et al, 1997). Cells were growing in complete media with 1 mM sodium pyruvate and 40 µM of DDC for 6 days. Media was replaced every second day and passaged if needed. After treatment cells were collected for experiment.

Where indicated, HeLa cells were treated with 0.5 µM Reversine (Sigma, R3904) or 50 nM Aphidicolin (Santa Cruz, sc-201535A) for 18 h, or pulsed for 4 h with 0.5 µM or 1 µM Camptothecin (CPT) (Selleckchem, NSC-100880) or 0.5 mM hydroxyurea (HU) (Sigma, H8627) followed by 18 h release. Alternatively, cells were treated with 0.5 µM Aphidicolin for up to 48 h, or pulsed with 1 µM CPT for four h and then released for 48 h. Where indicated, HeLa cells were pre-treated with 0.3 µg/mL Brefeldin A (Berger et al, 2017) for 30 min.

DiABZI (tlrl-diabzi-2) purchased from Invivogen, resuspended in nuclease-free water and used at the indicated working concentrations. DiABZI was added just prior to imaging. 2′3′ cGAMP (tlrl-nacga23-02) was purchased from Invivogen and resuspended in nuclease-free water with 25 mM HEPES pH 7.2–7.5 (ThermoFisher Scientific 15630-056).

All transgenic lines are available at request.

## Live cell confocal imaging

HeLa and HEK293T cells stably expressing the SIRF biosensor and H2B-mCherry were seeded in a µ-slide 8-well chamber precoated with ibiTreat (Ibidi). Xy positions were first predetermined and then treatments were added before live cell imaging experiment was initiated. Live cell imaging was performed using an automated Nikon Eclipse Ti2 inverted microscope equipped with a 20x dry objective (NA 0.75) or 40x dry objective (NA 0.95) and a Nikon DS-Qi2 high-sensitive CMOS monochrome camera. Multipoint acquisition was controlled by NIS-Elements 5.1 software. Image stacks were recorded every 1 h or 15 min for up to 18–24 h or 2 h, respectively, in an OkoLab environmental chamber at 37 °C and 5% $CO_2$. Alternatively, live cell imaging was performed using a fully motorized Nikon Ti2 with on stage incubation (temperature and $CO_2$) from OkoLab equipped with Crest X-Light V3 confocal scanning unit, Lumencor Celesta Light Engine and Andor Zyla-4.2P camera. Imaging was done with a 20x dry objective (NA 0.75).

For initial characterisation analyses (Figs. 1E,F, 2A,B, EV1F, EV2B,H) or complex stratification analyses (Figs. 3B,D, 6B, EV4, EV5). Images were analysed using ImageJ 2.0.0 software. In particular, biosensor fluorescence signal was tracked manually and monitored as median fluorescence insensitive (MFI). Relative signal was calculated by subtracting the background signal and dividing the intensity of all time points by the first frame. See "Biosensor automatic analysis with Cell Profiler" for additional high-throughput analyses.

## Live cell incubator imaging

For the apoptosis analyses in Fig. 5, HeLa SIRF biosensor cells were performed using IncuCyte S3 (Sartorius) at 37 °C 5% $CO_2$. For this, cells were seeded in a 96-well plate. Next day, cells were treated with either vehicle or BH-3 mimetic drugs ABT-737 (Hölzel;10 µM) and S63845 (10 µM) to induce MOMP/apoptosis in presence of pan-caspase inhibitor (QvD; 10 µM). STING agonist, diABZi was used as positive control for STING activation. After adding treatments, plate was inserted in IncuCyte chamber and 3 images per well were acquired every 1 h for 22 h. Images were analysed using IncuCyte analysis software module. Green positive object per $mm^2$, information was used to create the graph.

For the infection analyses in Fig. 4, HeLa biosensor cells were seeded at a density of five thousand cells per well in a 96-well plate. Next day, cells were infected with either (i) HSV-1 expressing mCherry (HSV1(17 +)Lox-CheP2AGLuc generously provided by B. Sodeik) (Devadas et al, 2014) or Influenza A (SC35M_NS1_2A_dsRED provided by M. Schwemmle) at the multiplicity of infection of 0.5. STING agonist, diABZI, was used as positive control for STING activation. One hour prior to diABZI treatment or HSV1 infection, 50 μM Acyclovir (MedChemExpress, HY-17422) was added where indicated. Circa half an hour after the cells were infected or treated, they were placed in an IncuCyte SX5 (Sartorius) at 37 °C, 5% $CO_2$ system, and imaged every 1–2 h for up to 3 days. Images were analysed using IncuCyte analysis software module.

For the titration of diABZI, HeLa biosensor cells were similarly seeded at a density of five thousand cells per well in a 96-well plate and treated with the following concentrations of diABZI: 0.05, 0.1, 0.5, 1 and 2 μM or with 0.5% DMSO for a control. The imaging was performed with IncuCyte SX5 (Sartorius) at 37 °C, 5% $CO_2$ system and cells were imaged every hour for up to 2 days.

Images were taken with 20x magnification and analysed in the Basic Analyzer mode quantifying the percentage of total green area (units in μm²/image). Parameters of IncuCyte® analyses: Green channel, Segmentation Top-Hat, Radius 100 μM, Threshold 2.3 GCU, Edge Split Off, Cleanup, Hole Fill 0, Adjust Size 1 px. Filters: Mean Intensity Min 1.2 μm²; Integrated Intensity: Min 18 μm²/Max 1200 μm².

## Biosensor automatic analysis with Cell Profiler

In Figs. 2D–H, 3E–G and EV2I, videos and images obtained from live cell and immunofluorescent imaging, respectively, where split into individual images containing a single colour channel and timepoint with the open-source programme BfConvert (https://docs.openmicroscopy.org/bio-formats/6.0.1/users/comlinetools/conversion.html). The individual images were then exported onto CellProfiler. For all other experiments, the H2B-mCherry nuclei would be identified and the borders of the nuclei would be expanded to match the borders of the HeLa cells for proper segmentation. Next, the biosensor signal was then identified and signal would be measured per single cell. The single-cell measurements were exported in a CSV file for further downstream analysis.

Data analyses scripts and other details for the SIRF biosensor quantifications using CellProfiler are further explain within the Github uploaded code "SIRF_Cell_Profiler.cpproj".

For the pCS2 + TOTO3 transfection experiment, cells were first segmented by identifying individual H2B-mCherry nuclei and discarding any with abnormally high intensities that might bleed into the Cy5 channel. The borders of the nuclei were then expanded to the edges of the HeLa cells to properly identify the individual cells. Next, the fluorescent plasmids were identified and HeLa cells with TOTO3 signal were categorised into a separate group. The biosensor signal was then identified and signal would be measured separately in transfected cells vs non-transfected cells. The single-cell measurements were exported in a CSV file for further downstream analysis. Details for the quantifications are explained in "Live_Cell_Imaging(40X_-Transfection).cpproj" (Github-uploaded code).

For all CSV files resulting from the Cell Profiler pipeline, R was used to either average the biosensor integrated intensity measured per individual cell in each condition or to calculate the percentage of cells containing biosensor signal vs the total cell population per image. Details for the R pipeline are shown in the Github uploaded code "SIRF_Cell_Profiler.R". For the pCS2 + TOTO3 experiment this was done individually for the transfected and non/transfected cells and described in the Github uploaded code "Live_Cell_Imaging(40X_Transfection).R".

## DNA and siRNA transfection

Cells were transfected with 25 nM or 50 nM siRNAs (Sigma) using Lipofectamine RNAiMAX transfection reagent (Thermo Fisher Scientific 13778075). 48–72 h after transfection, cells were further analysed. The following siRNAs were used: siRNA against Scramble (Sigma-Aldrich, SIC001) or an siRNA pool against firefly luciferase mRNA (U47296) (Dharmacon D-001206-14-20) as negative controls, an siRNA pool against TBK1 (Dharmacon M-003788-02-0010) and an siRNA pool against TREX1 (Dharmacon M-013239-03-0005).

For the circular plasmid transfection analyses, cells were transfected with 100 ng of pCS2+ containing SV40-mCherry per well of a μ-slide 8-well chamber precoated with ibiTreat (Ibidi) using Lipofectamine3000 transfection reagent immediately before imaging. Alternatively, biosensor cells were transfected with pCS2+ fluorescently labelled with Thiazole Red Homodimer (TOTO3) at concentrations the aforementioned concentrations right before imaging. Fluorescent labelling was done by mixing 5 μg of pCS2+ plasmid with TOTO3 at a ratio of one fluorophore per 10 nucleotides in 50 μl PBS. Reaction was carried out for an hour in the dark and fluorescently labelled plasmid was recovered via ethanol precipitation by adding 0.1 volumes of 3 M sodium acetate pH 5.2 and 3 volumes of pure ice-cold ethanol. Precipitation was done at −70 °C overnight, washed once with 75% cold ethanol and eluted in 20 μl of nuclease-free water.

For the seeding of transfected WT HEK293T cells on top of the HEK293T or HeLa biosensor monolayer this was first done by seeding WT HEK293T cells in a 6-well plate and on the next day transfecting either 1 μg SV40NLS-mCherry or cGAS-mCherry using Lipofectamine3000 as per manufacturer's instructions and incubating the cells for 24 h at 37 °C and 5% $CO_2$. After 24 h, the tranfected WT HEK293T cells were trypsinized and seeded onto biosensor cell line monolayer right before imaging

## Immunofluorescence

HeLa cells were seeded on coverslips in 12-well plates. Where indicated, cells were first transfected with siRNA for 48–72 h. Cells were fixed with 2% paraformaldehyde in PBS and stained, according to the experiment, for DAPI and the indicated proteins using rabbit anti-TGOLN2 (1:250, CST 55727) rabbit anti-IRF3 (1:500, CST 11904) rabbit anti-cGAS (1:200, CST 79979), mouse anti-Calnexin (1:250, Santa Cruz sc23954), mouse anti Lamin A/C (1:200, CST 4777), guinea pig anti-CENP-C (1:1000, MBL International PD030) and then probed with donkey anti-guinea pig Cy3 (1:800, Millipore AP193C), donkey anti-rabbit Alexa594 (ThermoFisher A21206) or donkey anti-rabbit Alexa647 (Thermofisher A31573).

Coverslips were imaged with a Nikon Eclipse Ti using a Nikon Plan Apo λ 60x NA 1.40 with oil immersion and the NIS Elements software. Data was analysed using ImageJ 2.0.0.

For the experiments in Fig. 4, cells were seeded on glass coverslips, treated with diABZI, vehicle or BH-3 mimetic drugs (S63845, Hölzel HY-100741 and ABT-747 Hölzel HY-50907) to induce MOMP/apoptosis, or BH-3 mimetic drugs with QvD (Hölzel, HY-12305g) fixed in 4% paraformaldehyde and blocked with BSA. Cells were immunostained for mouse anti-DNA (1:200, Progen-690014S) and rabbit anti-TOMM20 (1:200, Atlas Antibodies-HPA011562) and secondary antibodies used were Abberior STAR ORANGE and STAR-RED (Abberior, 1:200 diluted). Images were taken in the Abberior confocal microscope and processed in ImageJ.

## Western blotting

For Western blotting, cells were lysed in full lysis buffer (50 mM Tris-HCl, pH 7.5, 150 mM NaCl, 1% Triton X-100, 0.05% SDS, 1 mM β-mercaptoethanol, 2 mM EDTA, 1x protease phosphatase inhibitor cocktail (Thermo Fisher)). The cleared lysates were mixed with 4x NuPAGE loading buffer, resolved on 10% NuPAGE gels and transferred to nitrocellulose membranes. For western blot experiments, the following antibodies were used: mouse anti-α-tubulin (Sigma-Aldrich, T9026), rabbit anti-TBK1 (CST 38006), rabbit anti-STING (CST 13647), rabbit anti-phospho-TBK1 (CST 5483), rabbit anti-phospho-STING (CST 50907), rabbit anti-IRF3 (1:1000, CST 11904), cGAS (1:1000, CST 15102) and rabbit anti-TREX1 (CST 15107). Membranes were then probed with goat anti-mouse IgG HRP (Millipore AP308P) or goat anti-rabbit IgG HRP (CST 7074).

## Real-time PCR

RNA was extracted by using the RNeasy Mini Kit (QIAGEN), and the cDNA was synthesised from 1 μg of RNA (Bioline SensiFAST cDNA Synthesis Kit). Quantitative PCR was performed with the SensiFAST SYBR Hi-ROX Kit (Bioline) using a StepOnePlus 96-well plate reader (Applied Biosystems). For RT–qPCR data analysis, normalisation of gene expression was carried out to the house-keeping gene GAPDH.

| |
|---|
| IFNb Fw: 5′ CAACTTGCTTGGATTCCTACAAAG 3′ |
| IFNb Rev: 5′ TATTCAAGCCTCCCATTCAATTG 3′ |
| CXCL10 Fw: 5′ GGTGAGAAGAGATGTCTGAATCC 3′ |
| CXCL10 Rev: 5′ GTCCATCCTTGGAAGCACTGCA 3′ |
| IL-6 Fw: 5′ AGACAGCCACTCACCTCTTCAG 3′ |
| IL-6 Rev: 5′ TTCTGCCAGTGCCTCTTTGCTG 3′ |
| GAPDH Fw: 5′ TCAAGAAGGTGGTGAAGCAGG 3′ |
| GAPDH Rev: 5′ AGCAGGAAATGAGCTTGACAAA 3′ |
| Influenza A Fw: AGATGAGCCTTCTAACCGA |
| Influenza A Rev: GCAAAGACATCTTCAAGTCTC |

## Reporter assay

30000 WT HEK293T or HEK293T-SIRF biosensor cells were seeded in triplicate in a 96-well plate. Both HEK293T cells were transfected with 10 ng IFN-Beta_pGL3 (Addgene 102597), 5 ng Renilla, and up to a final amount of 50 ng with pCS2+. 1 ng WT

STING was added to WT HEK293T where indicated only. Transfection was done using XtremeGene-9 following the manufacturer's protocol Cells were then incubated for 24 h. Both HEK293T cells were then treated with the diABZI as described above for 24 h and then harvested.

## Image data processing

Raw images were imported to Fiji (ImageJ, v2.0) prior to their export to Photoshop 2020 for panel arrangement. Linear changes in contrast or brightness were equally applied to all controls and across the entire images. The models and schemes were created with BioRender.com.

## Statistical analyses

Data are shown as mean with standard error of the mean (SEM or SD), as indicated in the figure legends. Where indicated, Student's t-tests (two groups) or one-way ANOVA analyses with Tukey correction (three or more groups) were calculated using Prism v8. Significance is indicated as: $*P < 0.05$, $**P < 0.01$, $***P < 0.001$, or n.s.: not significant.

## Data availability

Source data for the imaging studies is available at BioImages: https://www.ebi.ac.uk/biostudies/bioimages/studies/S-BIAD1536. CellProfiler and R code is available at Github: Acebron-Lab/SIRF-Biosensor-Video-Processing https://github.com/Acebron-Lab/SIRF-Biosensor-Video-Processing.

The source data of this paper are collected in the following database record: biostudies:S-SCDT-10_1038-S44318-025-00370-y.

## Peer review information

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

## Acknowledgements

We thank A. Roers for sharing reagents, U. Engel for imaging support, and M. Mankarious for assistance in preliminary studies. We thank the Nikon Imaging Center and the FACS facility at the University of Heidelberg for access to microscopes, cytometers, and for technical help. This work was funded by the Deutsche Forschungsgemeinschaft (DFG, German Research Foundation)—SFB 1324—Project number 331351713 (Project B03 to SPA) and Heidelberg University through the Excellence Initiative – Explorer programme. SS holds a PhD fellowship from Boehringer Ingelheim Fonds. For the publication fee, we acknowledge financial support by Heidelberg University.

## Author contributions

**Steve Smarduch**: Data curation; Formal analysis; Validation; Investigation; Visualization. **Sergio David Moreno-Velasquez**: Conceptualization; Resources; Investigation; Methodology. **Doroteja Ilic**: Data curation; Formal analysis; Investigation. **Shashank Dadsena**: Resources; Data curation; Investigation. **Ryan Morant**: Validation; Investigation. **Anja Ciprinidis**: Formal analysis; Investigation. **Gislene Pereira**: Resources; Funding acquisition; Project administration. **Marco Binder**: Resources; Supervision. **Ana J García-Sáez**: Resources; Supervision. **Sergio P Acebrón**: Conceptualization; Formal analysis; Supervision; Funding acquisition; Investigation; Writing—original draft; Project administration.

Source data underlying figure panels in this paper may have individual authorship assigned. Where available, figure panel/source data authorship is listed in the following database record: biostudies:S-SCDT-10_1038-S44318-025-00370-y.

## Funding

## Disclosure and competing interests statement

The authors declare no competing interests.

# Expanded View Figures

**Figure EV1.   STING signalling analyses in HEK293T, HFF1 and HeLa cells.**

(**A**) PiggyBAC construct containing the SIRF biosensor. (**B**) Western blots of wt and SIRF biosensor HeLa cells treated as indicated. Note that the samples of HeLa biosensor cells correspond to the same experiment shown in Fig. 1C. (**C**) qRT-PCR analyses of wt and SIRF biosensor HeLa cells treated as indicated and represent as mean ± SD of 7–8 biological replicates. (**D, E**) Immunofluorescence analyses of wt HeLa cells treated as indicated. (**F**) Live cell confocal imaging analyses of HEK293T and HFF-1 cGAMP-biosensor cells treated with 2 μM diABZI. Data in the mean ± SEM of relative fluorescence (compared to t = 0) of 20 randomly selected cells per condition. (**G**) IFNb-Luc reporter assays in HEK293T cells transfected with empty plasmid or wt STING, or HEK293T cGAMP-biosensor cells, and treated with the indicated concentrations of the STING agonist diABZI, and represent as mean ± SD of 4 biological replicates. Note that HEK293T do not express endogenous cGAS or STING as shown in (**H**). (**H**) Western blots of wt HeK293T and HeLa cells. Scale bars = 10 μm. Source data are available online for this figure.

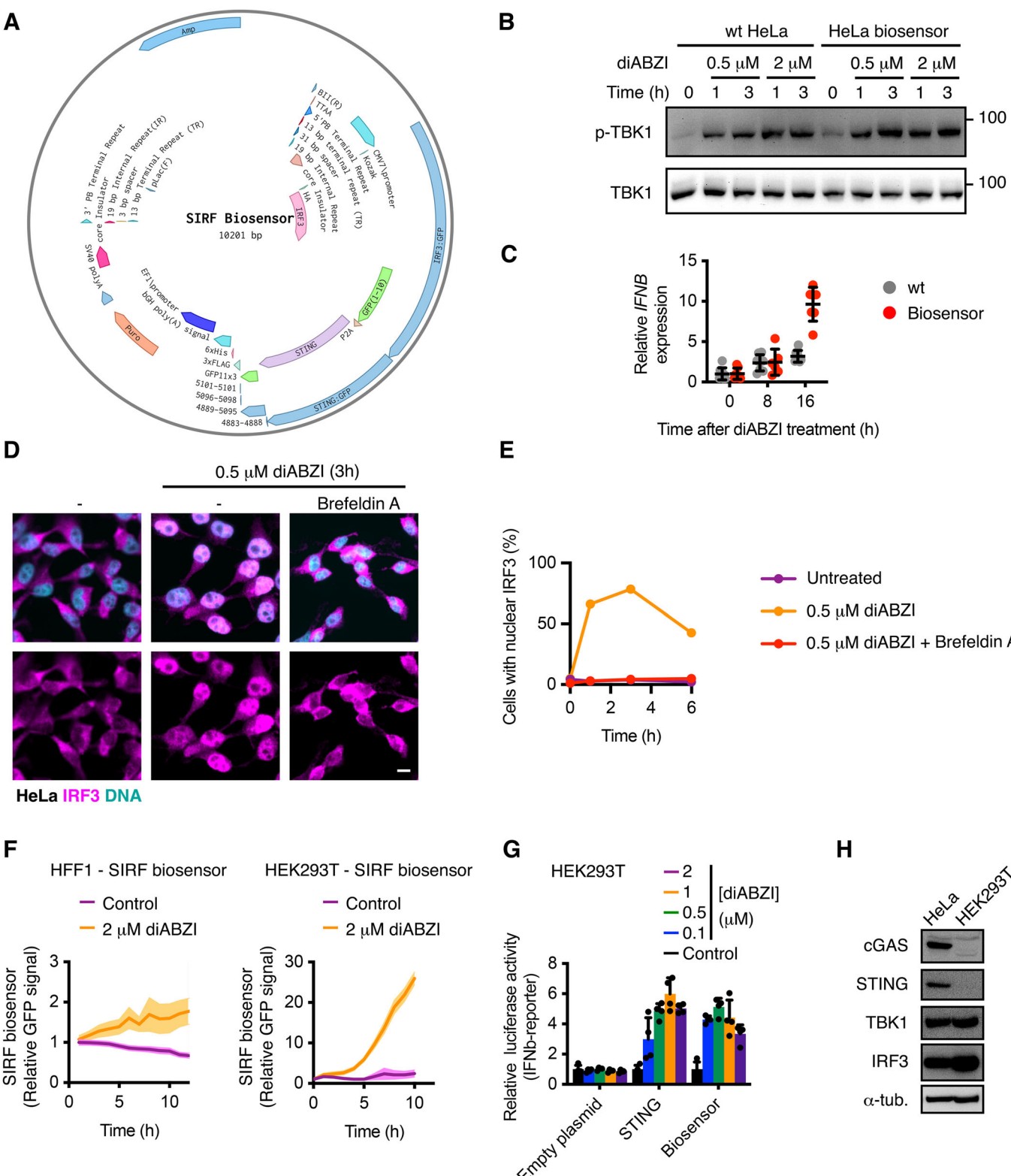

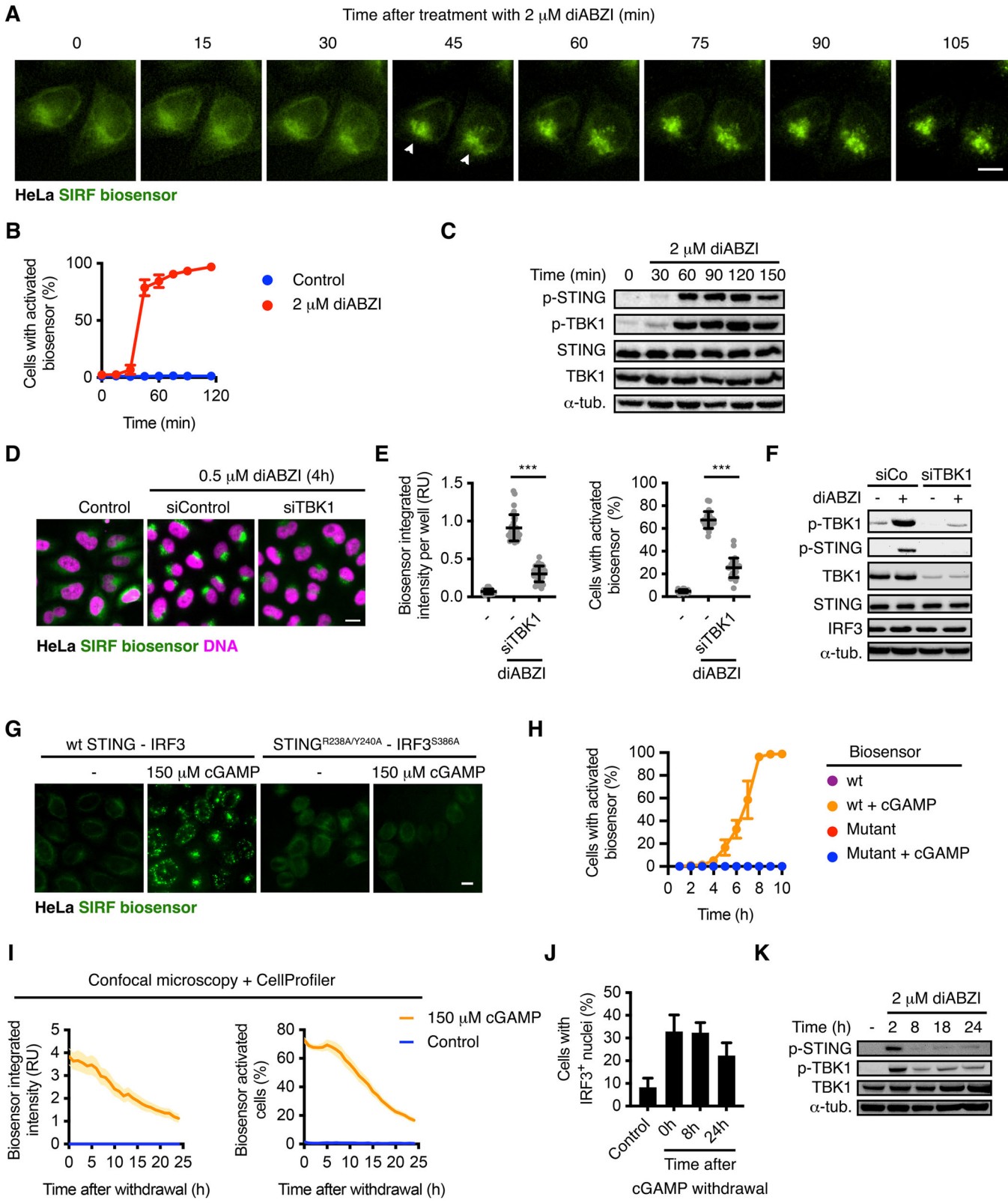

**Figure EV2.  SIRF biosensor activation dynamics and requirements.**

(A, B) Live cell confocal imaging analyses of HeLa SIRF biosensor cells in the presence or absence of 2 µM diABZI. In (A), arrows indicate the initial clustering of the biosensor. In (B), activation of the biosensor was scored as clustering of the GFP signal in 1 or more puncta, and it is represented as mean ± SEM of $N = 3$ independent time-lapses with >200 cells per condition. (C) Western blots of wt HeLa cells treated as in (A, B). (D, E) Immunofluorescence analyses of HeLa SIRF biosensor cells transfected with the indicated siRNAs and treated as indicated for 4 h. Note that cells were imaged in a 384x high content microscope prepared for screening and analysed using CellProfiler. Data represents mean ± SD of $N = 32$ different samples per condition, each analysed for % of active cells or integrated intensity (RU) using CellProfiler. In (E), $P$-values from one-way ANOVA between the indicated groups with $N = 32$ biological replicates are indicated as ***$P < 0.0001$ in both cases. (F) Western blots of HeLa cells treated as in (D, E). siCo, unrelated control siRNA siLRP6. (G, H) Live cell confocal imaging analyses of SIRF biosensor (wt) and STING(R238A/Y240A)-P2A-IRF3(S386A) (mutant) HeLa cells in the presence or absence of 150 µM cGAMP. In (G) activation of the biosensor was scored as clustering of the GFP signal in 1 or more puncta, and it is represented as mean ± SEM of $N = 3$ independent time-lapses with >200 cells per condition. (I) CellProfiler analyses of HeLa SIRF biosensor H2B-mCherry cells after live cell confocal imaging following cGAMP withdrawal (20-hour pre-treatment) or untreated. Data represent mean ± SEM of 8 different parallel time-lapses per condition each automatically analysed and with >200 cells. (J) Immunofluorescence analyses of wt HeLa cells after cGAMP withdrawal (20-hour pre-treatment) represented as mean ± SD of $N = 3$ biological replicates with >100 cells per condition (K) Western blots of wt HeLa cells treated as indicated. Scale bars = 10 µm. Source data are available online for this figure.

**A**

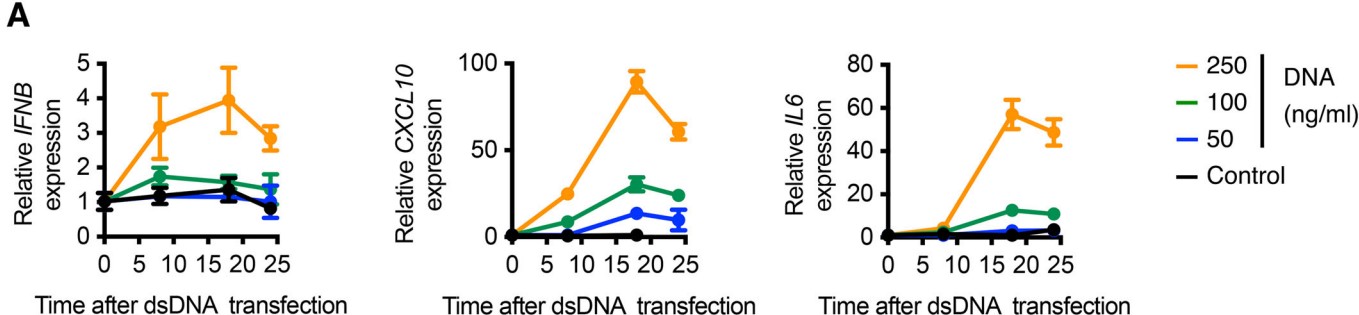

**B**

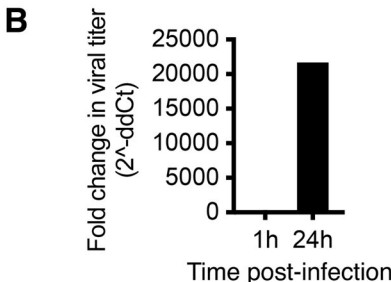

**Figure EV3.** **Innate immune response to foreign dsDNA and dsRNA in HeLa cells.**

(**A**) qRT-PCR analyses of wt HeLa cells treated as indicated and represented as mean ± SD of $N = 3$ biological replicates. (**B**) qRT-PCR analyses of influenza A viral RNA in infected HeLa SIRF biosensor cells, as indicated, and represented as mean of a representative experiment from $N = 3$ independent experiments. Source data are available online for this figure.

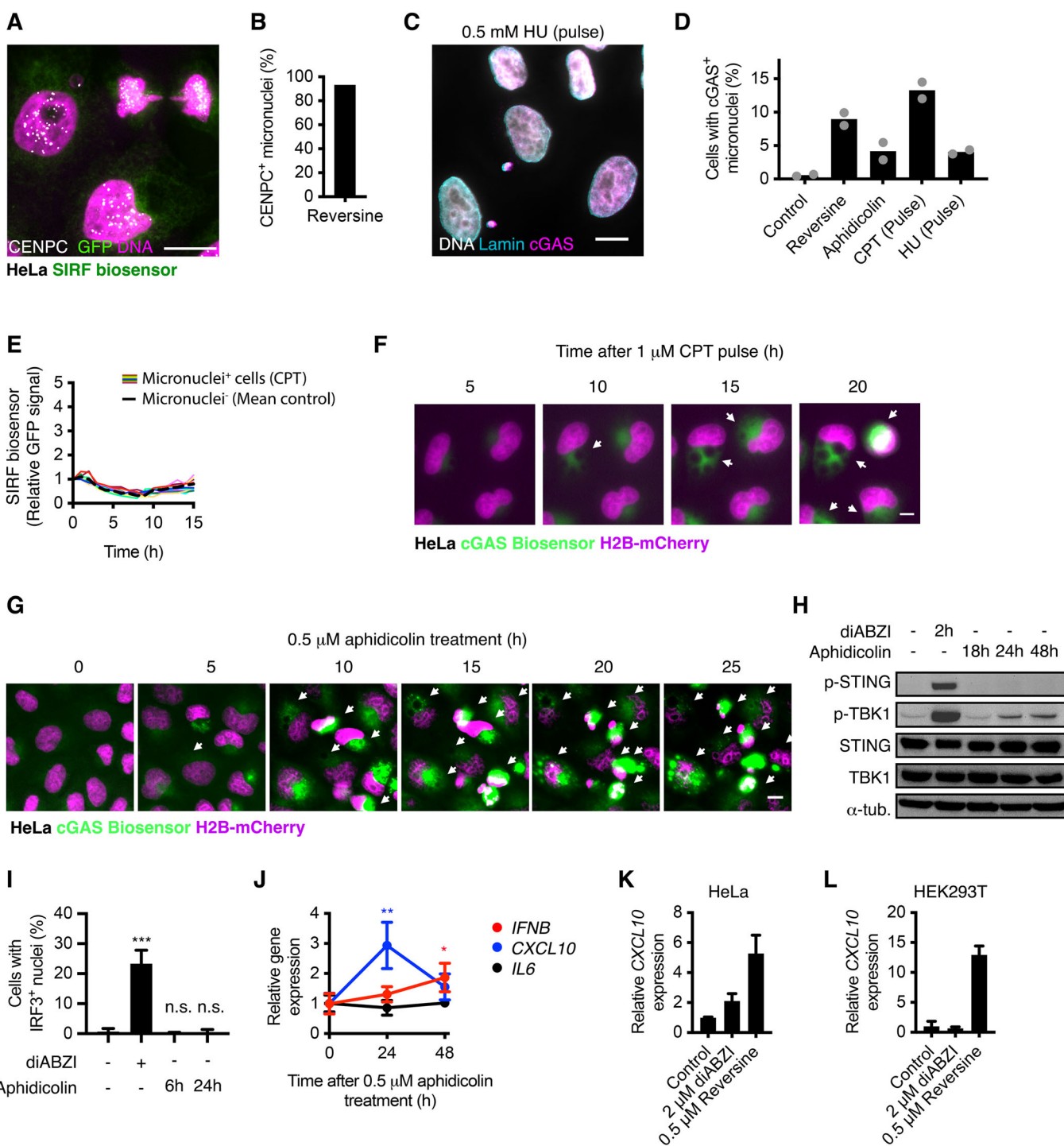

◀  **Figure EV4.  Characterisation of Micronuclei and replicative stress in HeLa cells.**

(**A, B**) Immunofluorescence experiments in HeLa SIRF biosensor cells treated with 0.5 µM reversine for 18 h and stained as indicated. (**C, D**) Immunofluorescence analyses of HeLa cells treated with 2 µM diABZI for 4 h, with 0.5 µM reversine or 50 nM aphidicolin for 18 h, or pulsed for 4 h with 0.5 µM CPT or 0.5 mM HU followed by 18 h release. CPT, Topoisomerase inhibitor Camptothecin; HU, Hydroxyurea. In (**D**), data is mean of $N = 2$ independent experiments each with >200 cells per condition. (**E, F**) Live cell imaging analyses of HeLa SIRF biosensor H2B-mCherry cells pre-treated with a pulse of 0.5 µM CPT for 4 h and released for 18 h. In (**E**), single cells with micronuclei at the beginning of the recording were monitored for 15 h and are compared to the mean of control cells (dashed line). In (**F**), cells showing clear signs of autophagy and cell death are indicated with arrows. (**G**) Live cell imaging analyses of HeLa SIRF biosensor H2B-mCherry cells treated with 0.5 µM aphidicolin (DNA polymerase inhibitor). Cells showing clear signs of autophagy and cell death are indicated with arrows. (**H**) Western blots of wt HeLa cells treated as indicated. (**I**) Immunofluorescence analyses of wt HeLa cells treated as indicated. Data represents mean ± SD of $N = 4$ biological replicates with *P*-values from one-way ANOVA between from the indicated groups are indicated as ***$P < 0.0001$, or not significant (n.s.) from left to right as $P = 0.983$, $P = 0.999$. (**J, K**) qRT-PCR analyses of wt HeLa cells treated as indicated. Data represents mean ± SD of $N = 3$ biological replicates. *P*-values from one-way ANOVA compared to control for each gene are indicated as **$P = 0.0087$ for *CXCL10* and *$P = 0.049$ for *IFNB*. In (**K**), cells were treated with diABZI for 4 h or with reversine for 24 h. (**L**) qRT-PCR analyses of wt HEK29T cells treated with diABZI for 4 h or with reversine for 48 h. Data represents mean ± SD of $N = 3$ biological replicates. Scale bars = 10 µm. Source data are available online for this figure.

          

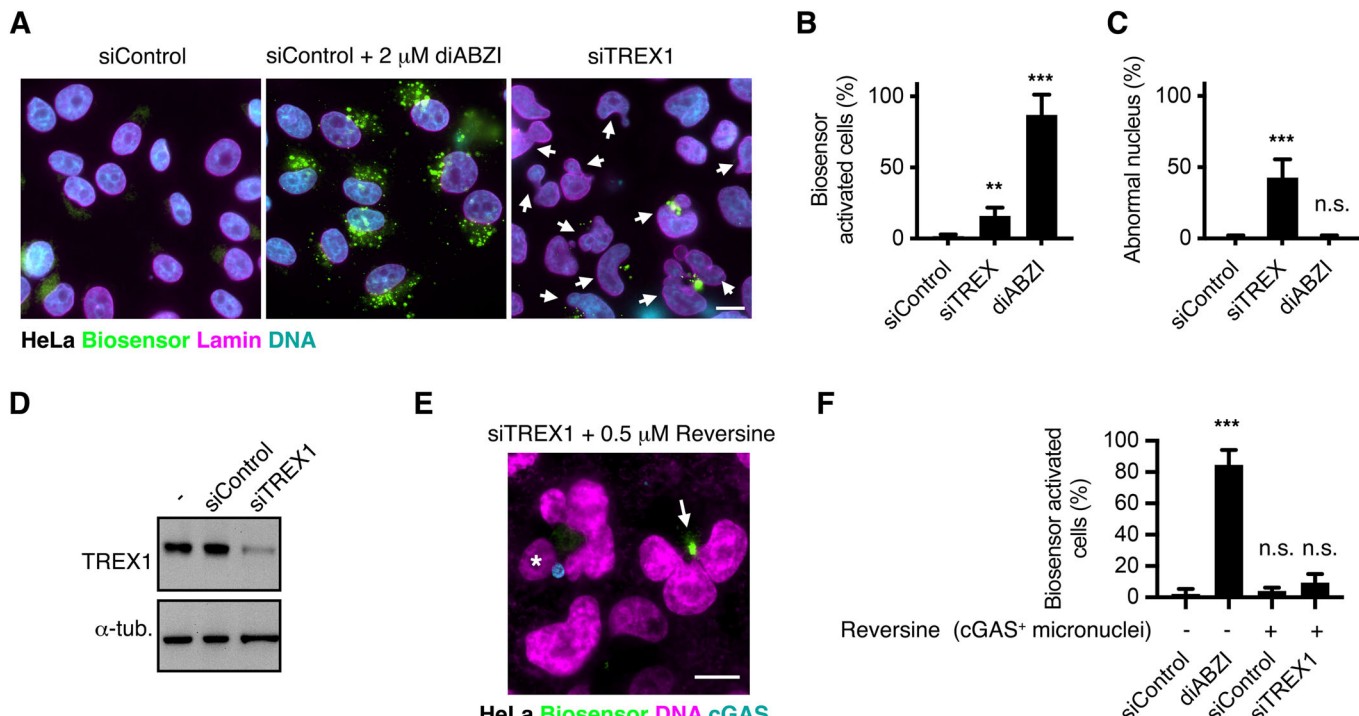

**Figure EV5.  SIRF biosensor activation upon TREX1 knock down.**

(A–C) Immunofluorescence analyses of HeLa SIRF biosensor cells transfected with the indicated siRNAs for 72 h, or treated with 2 μM diABZI for 4 h. In (C), abnormal nuclear shape was scored in cells showing similar morphology as those pointed in (A). Data represents mean ± SD of N = 8–10 biological replicates with at least 25 cells per sample. *P*-values from one-way ANOVA between the indicated groups are indicated in (B) as **P = 0.0035, and ***P < 0.0001, and in (C) as ***P < 0.0001 and n.s. = 0.9996. (D) Western blots of HeLa cells transfected as in (A–F). (E, F) Immunofluorescence analyses of HeLa SIRF biosensor cells transfected with the indicated siRNAs for 72 h, or treated with either 2 μM diABZI for 4 h or 0.5 μM reversin for 18 h. Note that reversine treated siTREX1 cGAS+ micronuclei cells (F) show less percentage of activation compared to cells only transfected with siTREX1 (B) suggesting that active biosensor in siTREX1 concentrate in cells with nuclear shape abnormalities and/or division failure. In (F), Data represents mean ± SD of N = 3 independent experiments. *P*-values from one-way ANOVA compared to control are indicated as ***P < 0.0001 and n.s. = 0.94 and 0.35, respectively. In (A, E) arrows indicate active biosensor upon siTREX1, while an asterisk marks a cGAS+ micronucleus. Scale bars = 10 μm. Source data are available online for this figure.

