## [Peer Review File · The EMBO Journal]

A novel biosensor for the spatiotemporal analysis of STING activation during innate immune responses to dsDNA

Steve Smarduch, Sergio Moreno-Velasquez, Doroteja Ilic, Shashank Dadsena, Ryan Morant, Anja Ciprianidis, Gislene Pereira, Marco Binder, Ana Garcia-Saez, and Sergio Acebron

Corresponding author(s): Sergio Acebron (sergio.acebron@cos.uni-heidelberg.de)

Review Timeline:

Submission Date:	21st Jun 24
Editorial Decision:	16th Jul 24
Revision Received:	16th Nov 24
Editorial Decision:	9th Dec 24
Revision Received:	23rd Dec 24
Accepted:	10th Jan 25

Editor: Ioannis Papaioannou

Transaction Report:

Dear Sergio,

Thank you again for submitting your manuscript EMBOJ-2024-118005 for consideration by The EMBO Journal and for your patience during peer review. Your manuscript has now been seen by three experts in the field, and we have received the full set of their comments, which you can find below.

As you will see, all referees recognize the novelty of the biosensor, as well as its potential usefulness and applicability in various cellular settings to monitor the dynamics of the cGAS-STING pathway activation. They also appreciate the overall quality of the work, and they are supportive of publication of this work in The EMBO Journal, provided that a number of concerns they raise will be adequately addressed in a revised version of the manuscript. In particular, they point out that the biosensor responds to STING activation regardless of the nature of the upstream signals (instead of being *sensu stricto* a cGAMP-specific sensor), and we think that this should be fully clarified in the revised manuscript. They also list a number of suggestions that together call for the need for more clarity on the mechanisms underlying micronuclei interaction with the innate immune response. They also provide other useful suggestions for the improvement of the presentation of the Results and the Figures, which should be taken on board. Finally, we think that any additional data on a proof-of-principle screen using this biosensor and/or additional cell models/organoids -which you have mentioned before that you have been working on- would significantly benefit the manuscript and increase its impact on the field.

Given the referees' comments and recommendations, as well as your expressed willingness to expand the study further with additional data, I would like to invite you to submit a revised version of the manuscript along with a detailed point-by-point response addressing all referees' comments. I should add that it is The EMBO Journal policy to allow only a single round of major revision, and acceptance of your manuscript will therefore depend on the completeness of your responses in this revised version. Please let me know if you have any questions or comments that you would like to discuss with me.

We generally allow three months as standard revision time (October 15, 2024). As a matter of policy, competing manuscripts published during this period will not negatively impact our assessment of the conceptual advance presented by your study. However, we request that you contact us as soon as possible upon publication of any related work, to discuss how to proceed. Should you foresee a problem in meeting this three-month deadline, please let us know in advance and we may be able to grant an extension.

Thank you again for the opportunity to consider your work for publication in The EMBO Journal. I am looking forward to your revision.

Best regards,

Ioannis

Instructions for preparing your revised manuscript

1. When you are ready to submit the revision, please upload:

- A Word file of the manuscript text (including legends of main Figures, EV Figures and Tables). Please make sure that changes are highlighted (or "tracked") to be clearly visible.

- Individual production-quality figure files (one file per figure). When assembling your figures, please refer to our figure preparation guidelines in order to ensure proper formatting and readability in print as well as on screen:

If the data shown in a figure are obtained from n {less than or equal to} 2, please use scatter plots showing the individual data points.

i. the name of the statistical test used to generate error bars and P values

ii. the number (n) of independent experiments (please specify technical or biological replicates) underlying each data point

(discussion of statistical methodology can be reported in the Materials and Methods section, but figure legends should contain a basic description of n, P, and the test applied)

iii. the nature of the bars and error bars (s.d., s.e.m.).

- A point-by-point response to the referees' comments, with a detailed description of the changes made (as a word file). All referees' concerns must be fully addressed and their suggestions taken on board. When preparing your letter of response to the referees' comments, please bear in mind that this will form part of the Review Process File and will therefore be available online to the community. Please note that you have the possibility to opt out of the transparent process at any stage prior to publication by letting the editorial office know (contact@embojournal.org); if you do opt out, the Review Process File link will point to the following statement: "No Peer Review File is available with this article, as the authors have chosen not to make the review process public in this case.". For more details on our Transparent Editorial Process, please visit our website: <https://www.embopress.org/page/journal/14602075/authorguide#transparentprocess>

- Expanded View (EV) files (replacing Supplementary Information) that are collapsible/expandable online. A maximum of 5 EV Figures can be typeset. EV Figures should be cited as "Figure EV1, Figure EV2" etc. in the text, and their respective legends should be included in the manuscript file after the legends of regular figures. See detailed instructions regarding Expanded View files here:

- For the figures that you do NOT wish to display as Expanded View figures, they should be bundled together with their legends in a single PDF file called "Appendix", which should start with a short Table of Contents (including page numbers). Appendix figures should be referred to in the main text as: "Appendix Figure S1, Appendix Figure S2" etc. Please see detailed instructions here: <https://www.embopress.org/page/journal/14602075/authorguide#expandedview>

- A complete author checklist, which you can download from our author guidelines (<https://www.embopress.org/page/journal/14602075/authorguide>). Please note that the checklist will also be part of the Review Process File.

2. Please note that no statistics should be calculated and shown in Figures if $n=2$. Please also note that each p value should be reported as an exact value.

3. Before submitting your revision, primary datasets (and computer code, where appropriate) produced in this study need to be deposited in appropriate public databases (see <https://www.embopress.org/page/journal/14602075/authorguide#dataavailability>). The accession numbers, databases, and the specific URLs (links) should be listed in a formal "Data availability" section (placed after Materials and Methods) that follows the model below (see also <https://www.embopress.org/page/journal/14602075/authorguide#dataavailability>):

Data availability

- RNA-seq data: Gene Expression Omnibus GSE46843 (<https://www.ncbi.nlm.nih.gov/geo/query/acc.cgi?acc=GSE46843>)
- [data type]: [name of the resource] [accession number/identifier/doi] ([URL or identifiers.org/DATABASE:ACCESSION])

*** All links should resolve to a page where the data can be accessed. ***

*** Please remember to provide in the Data availability section of your revised manuscript reviewer passwords if the datasets are not yet public. ***

*** The Data Availability Section is restricted to new primary data that are part of this study. In case you have no data that require deposition in a public database, please state so instead of referring to the database: "Our study includes no data deposited in public repositories." under the heading "Data availability". ***

4. Please check that the title and the abstract of the manuscript are brief, yet explicit, even to non-specialists. The length of the title should not exceed 100 characters, and the abstract should be a single paragraph not exceeding 175 words.

5. The Materials and Methods need to be described in the manuscript using our "Structured Methods" format, which is now required for all research articles. According to this format, the Materials and Methods section includes a single "Reagents and Tools Table" -listing key reagents, experimental models, software and relevant equipment and including their sources and relevant identifiers- followed by a "Methods and Protocols" section describing the methods. More information on this format as well as detailed instructions, examples, and a template (.docx) for the "Reagents and Tools Table" can be found in our author guide: <https://www.embopress.org/page/journal/14602075/authorguide#structuredmethods>.

6. Please also note our reference format: <https://www.embopress.org/page/journal/14602075/authorguide#referencesformat>.
7. At EMBO Press we ask authors to provide source data for the main manuscript figures. Our source data coordinator will contact you to discuss which figure panels we would need source data for and will also provide you with helpful tips on how to upload and organize the files.
8. Please remember: digital image enhancement is acceptable practice, as long as it accurately represents the original data and conforms to community standards. If a figure has been subjected to significant electronic manipulation, this must be noted in the figure legend or in the "Materials and Methods" section. The editors reserve the right to request original versions of figures and the original images that were used to assemble the figure.
9. Our journal encourages inclusion of data citations in the reference list to directly cite datasets that were obtained from public databases. Data citations in the article text are distinct from normal bibliographical citations and should directly link to the database records from which the data can be accessed. In the main text, data citations are formatted as follows: "Data ref: Smith et al, 2001" or "Data ref: NCBI Sequence Read Archive PRJNA342805, 2017". In the Reference list, data citations must be labeled with "[DATASET]". A data reference must provide the database name, accession number/identifiers, and a resolvable link to the landing page from which the data can be accessed at the end of the reference. Further instructions are available at: <https://www.embopress.org/page/journal/14602075/authorguide#referencesformat>.
10. We request authors to consider both actual and perceived competing interests. Please review our policy (<https://www.embopress.org/page/journal/14602075/authorguide#conflictsofinterest>) and update your competing interests statement if necessary. Please name this section 'Disclosure and competing interests statement' and place it after the Acknowledgements section.
11. Please note that all corresponding authors are required to provide an ORCID ID upon submission of a revised manuscript (<https://orcid.org/>). Please find instructions on how to link your ORCID ID to your account in our manuscript tracking system in our Author guidelines (<https://www.embopress.org/page/journal/14602075/authorguide#authorshipguidelines>).
12. We use CRediT to specify the contributions of each author in the journal submission system. CRediT replaces the author contribution section, which should be removed from the manuscript. Please use the free text box to provide more detailed descriptions. See also guide to authors: <https://www.embopress.org/page/journal/14602075/authorguide#authorshipguidelines>.
13. Further information is available in our Guide For Authors: <https://www.embopress.org/page/journal/14602075/authorguide>
14. We would also welcome the submission of cover suggestions or motifs to be used by our Graphics Illustrator in designing a cover.
15. Please use the link below to submit your revision:
<https://emboj.msubmit.net/cgi-bin/main.plex>

Referee #1:

In the manuscript, the authors reported a novel probe to monitor the activation of cGAS-STING pathway, which they call a cGAMP sensor. They examined this novel probe in several different cellular settings, such as dsDNA-virus infection, mitochondria-mediated apoptosis, and chromosomal missegregation/micronuclei formation. The experiments are well designed, and the data are generally convincing.

The authors might find the following specific comments useful to improve the study.

Specific comments:

(1)
Figure 1D/1H: The images are blurry, compared to others such as Figure 2D. It should be corrected.
Figure 1G: The markers for trans-Golgi and ER should be described.

(2)
The authors should examine extensively the cGAS-STING activation with genome DNA leakage. A number of previous papers

showed the activation of cGAS-STING pathway in various types of cells under DNA replication stress, nuclear membrane rupture, or dysfunction of cytoplasmic DNase (Trex): these stresses may not be necessarily linked to micronuclei formation. Activation of the probe in one of these settings should greatly enhance the interest of the probe in the research of pathogenesis of various autoinflammatory diseases, thus this issue is critical.

This reviewer appreciates the results in Figure 5 demonstrating no STING activation upon micronuclei formation, which supports the recent papers (Takaki et al., 2024 Mol Cell; Sato et al., Life Cell Alliance 2024). Figure 5 can remain as it is.

Minor comments:

(3) Two papers (Kuchitsu et al., Nat Cell Biol 25, 453-466 (2023); Xun et al., EMBO Rep 25, 544-569 (2024)) should be cited in the context of the role of endolysosomal trafficking of STING in its signalling termination.

(4) As described above, the paper by Sato should be cited, along with Takaki et al.

(5) This reviewer is wondering if the term "cGAMP sensor" is adequate, because the probe does not quantitate the amount of cGAMP.

Referee #2:

The authors report to develop a sensor for cGAMP levels, to be used as a readout for the activity of the innate inflammatory pathway signaling through TBK1 and IRF3. We find that this work represents an interesting and noteworthy attempt to develop the necessary single cell approaches aimed at monitoring the heterogeneity of cellular responses. The authors show that the biosensor is responsive to multiple sources of stimulation. We would support the publication of this manuscript in case the following comments are addressed.

Major comments

1. The authors state that the tool is a sensor for cGAMP levels, however it is a monitor for STING activation. It is well established that there are alternative ways of STING activation, independent of cGAMP, and there is no reason to suspect that the sensor would not respond to such stimuli. To confirm the contribution of cGAMP to STING activation, the authors would have to mutate the cGAMP-ligand binding domain of the sensor. We suggest that the manuscript's title and content is adjusted to clarify that this tool measures activation of STING, irrespective of the upstream inputs. The manuscript even shows that upon prolonged CPT treatment which leads to autophagy, the biosensor gets activated (Fig. S3F), which is in line with all the literature on the role of STING activation in autophagy.

2. Throughout the manuscript, it is unclear how many independent experiments are performed in each dataset/figure. Please specify and repeat where necessary.

3. The authors write about the ability of the sensor to report on the dynamics of the activation of the pathway. However this claim is not well substantiated, since they do not characterize the reversibility of the activation. They report that the washout of the STING agonist leads to loss of fluorescence, but this seems to be a slow response and a suboptimal setting. We suggest that this may be better addressed in response to cGAMP treatment. In fact, it is rather concerning/surprising the GFP levels continue to rise for hours on end (up to 10h of STING agonist treatment), while prior literature has demonstrated that once stimulated, the pathway also needs to be switched off, which is driven at least in part by STING degradation. Does the sensor follow this same trend? Are the kinetics of activation/inactivation coinciding with the kinetics of activation/inactivation of endogenous STING? We find that it is important to better characterize the (ir)reversibility of the sensor's activation.

4. When characterizing a new system, it is best to have a good reference point. Takaki et al. recently nicely showed IRF3 translocation to be a very sensitive readout for the activity of the pathway. Therefore, we suggest that the figures 1D-E show not only the imaging of the newly developed reporter system but to also show how this compares to IRF3 translocation. We find that further characterization here is also necessary through additional approaches. It would be nice if authors could show that interaction of the tagged STING and IRF3 occurs through a method other than measuring GFP fluorescence. This could be done by PLA or IP but also prevented by brefeldin A. Of course, to assess the contribution of the sensor to the activation, these experiments would be most easily interpretable and designed in the cellular background where no STING or IRF3 is expressed (unlike the HeLa cells the authors establish the sensor in).

5. To explain the findings in Fig.3 (namely, that many mCherry- cells show activation of the reporter), the authors propose a model of paracrine spreading of signal. What caught our eye is that the highlighted cell in panel A of Fig. 3 which is mCherry- and shows activation of the pathway even before the first transfected responder cells, is found far away from any responding

cells. The authors mention that they find no distance-dependency when it comes to this 'bystander' activation while one would expect paracrine signals to act in a distance-dependent manner. We think this model and data needs further investigation, before strong conclusions on paracrine signalling can be drawn.

6. Further, we find the result in Fig. 5J to be very surprising, as it shows that the response is much higher in the cells where the direct stimulus is not present (paracrine stimulation is higher than the direct transfection effects). The authors try to show that the biosensor is responsive to multiple sources of stimulation. However, with such unexpected results in this experiment, we think it would be important to do a more careful characterization of the system's responsiveness to exogenous cGAMP (limited data shown in Fig.1I), similar to the characterizations in Fig.5.

7. The data on the activation of the sensor gained in response to more physiological stimulation would be beneficial. Most of the experiments show that the system responds well to a very strong STING agonist (again, with limited data following stimulation with cGAMP shown in Fig.1I).

Minor comments

1. As previously mentioned in one of our major comments, it is unclear which part of the activation of the pathway can be attributed to the biosensor and which is based on the endogenously-expressed STING and IRF3. We find that the WB in Fig.1C is mislabeled, as the authors indicate that they detect GFP-tagged portions of STING when both endogenous and 'biosensor' pools of STING would be detected by a p-STING antibody. It is unclear what the 2 bands of the pSTING part of the blot represent. The authors need to clarify which band is which. Also, we suggest blotting for IRF3 levels in Fig. 1C. This would give a better description of the reporter system.

2. Figure 3H, please show that Influenza A sequence is present in cells.

3. Figure S3G, please show TREX1 levels on western blot upon siTREX1.

4. Please describe in materials and methods how the treatments with cGAMP are performed. How is the mock sample treated?

5. On page 7 of the manuscript the authors write that micronuclei not activating the pathway is unexpected. Please refer to Flynn et al, 2021 here. This is the first paper (to our knowledge) to report that micronuclei are not (strong) activators of STING.

6. Please show that HEK293T cells do not express cGAS and STING. (fig. S2C)

7. In Fig. 3A at timepoint 20h, the cells appear to be very unhappy. It is difficult to see since the only channel shown is GFP. In any case, we think it is important to show that the stimulation of the biosensor is not toxic.

8. Fig. 4C does not have a QvD only condition.

Referee #3:

In this work, Smarduch et al. generate a novel biosensor for cGAMP detection in cells using a split GFP system tagged to STING and IRF3. The authors show that this system, when stably expressed in cells, can be activated by STING agonists (synthetic and cGAMP), viruses, DNA (with cGAS activation), mitochondrial DNA upon apoptosis, but not in the context of micronuclei. Overall the authors provide convincing evidence that this system would be extremely useful to easily monitor the live propagation of cGAMP between cells in tissues. This tool should represent a very useful solution to the monitoring of cGAMP dynamics, however a few points need to be addressed to broaden its utility.

Major points:

1- The authors rely heavily on two aspects of their biosensors; on one hand, the biosensor forms intracellular aggregates/punctuates, as previously reported; On another these have increased fluorescence compared to background fluorescence. What is not very clear is what is used in which figures and how easily this is assessed. There is little information about how many cells were counted for the MFI (200 seems to be the cut of for punctate positive cells) - but the methods suggest a lot of this was done manually.

The authors seem to switch between one and the other in their figures - which is a bit confusing (although this reviewer understands that the authors are trying to show different aspects of the system). If the cGAMP GFP intensity can indeed be accurately measured and is increased upon stimulation, then the assay should also be validated on flow cytometry for a true quantification of fluorescence at the single cell level (which would be unbiased and more accurate than the user selected cells measured by microscopy). The abstract suggests that the biosensor can be used for plate reader (presumably whole well assay in fluorescent plate reader) - this should also be demonstrated somewhere. Ultimately, the authors need to better support the

single cell level approach using unbiased analyses.

2- One of the key aspects of this research is the capacity afforded by this technology to assess the propagation of cGAMP within tissue. The authors touched on this but do not address the capacity of cGAMP to propagate through gap junctions. Since HEK 293T cells form CX43 dependent gap-junctions, it would be easy to confirm this with minimal experiments. For example, co-culture of a few HEK 293T cells overexpressing cGAS-GFP (which make cGAMP) added on top of a confluent layer of HEK293T cells with the STING-IRF3 biosensor would allow to measure activation of the signal as it progresses through the cell monolayer.

3- The authors propose that micronuclei do not activate their biosensor. Because the micronuclei function/sensing would be expected to be very different between cell types it is important to tone-down the fact that this may be the case in their model - i.e. Hela cells - but maybe not in other models. One point which should be definitely be mentioned is the case of non-canonical STING signaling, which does not depend on cGAS and thus would not be able to recruit the biosensor. In this regard, it would be very useful to create stable HaCaT cells expressing this biosensor and show that while diABZI activate the biosensor, DNA damage with CPT/ does not (see Dunphy et al., Al-Asmari et al.), although it activates production of IL-6. This would complete the characterisation of this reporter and nuance the findings on sensing of DNA damage products presented here.

4- The sensitivity of the assay should be determined using fluorescent labelled dsDNA (e.g. ISD70), transfected in cells and biosensor activity in transfected cells measured with time. Unlike the DNA transfection used here (noting that the authors used SV40-mcherry which may increase the amount of cytosolic vector if the vector has an SV40 ori), transfection of defined amount of labelled synthetic DNA will allow to define a molecular concentration of cytosolic DNA sensed. This should be compared to the induction of IRF3 targets by RTqPCR to similar amount of transfected DNA in Hela cells.

Minor points:

- the text refers to Figure S1C but this should be S2 (in results section).
- the abbreviation for interferon is IFN not INF (which is used throughout the paper)
- Why do they use SV40-mcherry? - noting that SV40 has an impact on sensitization of cells to cGAS sensing by DNA damage (Pepin et al, mBio 2018).

Dear Referees,

Since the initial review of the manuscript, we have comprehensively addressed the issues raised by you. A detailed point-by-point response to all comments follows below. We now provide additional experimental evidence (New Figures 1C, 2D-H, 3A-G, 4A-D, new expanded view (EV) Figures EV1B-E,H, EV2E,G-K, EV3, EV4G-L, EV5, Movies S5, S6, S10, and referee figures 1-4), which further expands on the relevance of our biosensor to monitor the cGAS-STING response, and addresses standing question regarding the innate immune response to DNA damage and micronuclei. Furthermore, we now include additional scripts/approaches to utilise the biosensor for high-throughput imaging.

Thank you for your critical input, which we believe has substantially improved our manuscript.

(Please note that Supplementary [expanded view] figures are not merged with the main text and figures, and instead presented as single files, as per journal's guidelines for the revision)

Referee #1:

In the manuscript, the authors reported a novel probe to monitor the activation of cGAS-STING pathway, which they call a cGAMP sensor. They examined this novel probe in several different cellular settings, such as dsDNA-virus infection, mitochondria-mediated apoptosis, and chromosomal missegregation/micronuclei formation. The experiments are well designed, and the data are generally convincing.

Thank you

The authors might find the following specific comments useful to improve the study.

Specific comments:

(1)

Figure 1D/1H: The images are blurry, compared to others such as Figure 2D. It should be corrected.

We have edited Figure 1D and substituted Figure 1H with new cGAMP titrations and analyses shown in Figure 2G.

Figure 1G: The markers for trans-Golgi and ER should be described.

We now indicate the markers (TGOLN2 for Trans-Golgi and calnexin for ER) in the figure.

(2)

The authors should examine extensively the cGAS-STING activation with genome DNA leakage. A number of previous papers showed the activation of cGAS-STING pathway in various types of cells under DNA replication stress, nuclear membrane rupture, or dysfunction of cytoplasmic DNase (Trex): these stresses may not be necessarily linked to micronuclei formation. Activation of the probe in one of these settings should greatly enhance the interest of the probe in the research of pathogenesis of various autoinflammatory diseases, thus this issue is critical.

We addressed this point through several experiments and by revisiting the existing literature:

Previous literature: We have revisited previous studies linking gDNA to STING activation, and noticed that in most cases the readout for cGAS-STING were i) cGAS recruitment to gDNA (Flynn *et al*, 2021; Marinello *et al*, 2022; Martin *et al*, 2024; Mohr *et al*, 2021), which as shown by us (Figure 6 and EV5) and contemporaneous studies (Sato & Hayashi, 2024; Takaki *et al*, 2024) is not sufficient to activate STING, or ii) expression analyses of interferon and other cytokines which can be regulated by various innate immune response pathways (Deng *et al*, 2014; Harding *et al*, 2017; Hong *et al*, 2022; Marinello *et al.*, 2022; Mohr *et al.*, 2021), but iii) not STING activation (e.g., phospho-STING).

1) Replication stress: Our previous analyses of mild replication stress using 50 nM aphidicolin did not revealed activation of the biosensor or p-STING (Fig 6G-I). We induced severe replication stress using 500 nM aphidicolin, which resulted in increased fluorescence (albeit without the characteristic biosensor clustering) in dying HeLa cells within 5-20 hours (Figure EV5G), possibly due to STING involvement in autophagy (Gui *et al*, 2019) (See also referee #2, question #1). Parallel WB, IF, and qPCR analyses did not reveal phospho-STING or nuclear IRF3 upon 500 nM aphidicolin (Fig. EV5H,I). However, we detected increased expression of *IFNB* and *CXCL10* (Fig. EV5J, further explained below). These results, together with our previous data on CPT (Figure EV5F and 6G-I), suggest that replication stress does not directly trigger STING activation in HeLa cells. Intriguingly, continuous treatment with 500 nM aphidicolin slightly increased phospho-TBK1, which could account for the activation of other TBK1-related pattern recognition pathway (Li & Wu, 2021), and explain the increased *IFNB* and *CXCL10* expression. Indeed, we now show that reversine induces *CXCL10* expression even in HEK293T cells (Fig. EV5K,L), which lack cGAS (Fig. EV1G,H). Given that both aphidicolin and CPT treatments trigger cell death / autophagy, further mechanistic studies would be required to untangle the contribution of gDNA, mtDNA (through apoptosis), and other cascades (e.g., dsRNA) to the innate immune response under replicative stress.

2) Nuclear envelope integrity: We knocked down LaminA/C in HeLa cells, which did not activate the biosensor within 48 hours, but led to both cell death and biosensor activation after 72 hours. It is unclear to us whether these results are HeLa cell specific, but we have included this data for your perusal as Referee Figure 1. Of note recent evidence indicate that loss of nuclear envelope integrity does not necessarily activate STING (En *et al*, 2024).

Referee Figure 1: Knock-down of LaminA/C leads to cell death in HeLa after 72h

3) TREX1: We noticed that knock down of TREX1 resulted in i) severe nuclear shape abnormalities in 43% of HeLa cells, possibly due to its roles in protecting the nuclear envelope (Nader *et al*, 2021) and ii) activation of the biosensor in 16% of HeLa cells (Figure EV5A-D). These results suggest that the biosensor can be used to monitor TREX1-

associated auto-inflammatory pathologies, in addition to those associated with mtDNA (Figure 5). Of note, reversine treated siTREX1 HeLa cells harbouring cGAS+ micronuclei displayed lower percentage of activation compared to cells only transfected with siTREX1 (Fig. EV5B vs EV5F, which are obtained from the same set of experiments). This could be caused by a failure to divide of siTREX1 cells with abnormal nuclei and active STING (Fig. EV5A-C). Indeed, we detected a 2-fold reduction of micronuclei positive cells in siTREX1 vs siControl HeLa cells treated with reversine across our 3 independent experiments. As such, we believe that the functional interaction between STING-TREX1 with gDNA requires further investigation, and it is outside of the scope of this manuscript.

We now further discuss in the manuscript the relevance of STING to study TREX1 deficiency. We thank the referee for the suggestion.

This reviewer appreciates the results in Figure 5 demonstrating no STING activation upon micronuclei formation, which supports the recent papers (Takaki et al., 2024 Mol Cell; Sato et al., Life Cell Alliance 2024). Figure 5 can remain as it is.

Thank you

Minor comments:

(3) Two papers (Kuchitsu et al., Nat Cell Biol 25, 453-466 (2023); Xun et al., EMBO Rep 25, 544-569 (2024)) should be cited in the context of the role of endolysosomal trafficking of STING in its signalling termination.

We include them now in the introduction, as well as in the results section related to cGAMP withdrawal and autophagy (Page 4,9)

(4) As described above, the paper by Sato should be cited, along with Takaki et al.

Done

(5) This reviewer is wondering if the term "cGAMP sensor" is adequate, because the probe does not quantitate the amount of cGAMP.

After taking in consideration the feedback of the 3 reviewers, we decided to name it "SIRF (STING-IRF3) biosensor" along the text. We also include titrations of diABZI and cGAMP, and a pipeline for automatic analysis of the biosensor, which showcase the versatility of this tool to monitor STING agonists at different concentrations (Figure 2).

Referee #2:

The authors report to develop a sensor for cGAMP levels, to be used as a readout for the activity of the innate inflammatory pathway signaling through TBK1 and IRF3. We find that this work represents an interesting and noteworthy attempt to develop the necessary single cell approaches aimed at monitoring the heterogeneity of cellular responses. The authors show that the biosensor is responsive to multiple sources of stimulation. We would support the publication of this manuscript in case the following comments are addressed.

Thank you

Major comments

1. The authors state that the tool is a sensor for cGAMP levels, however it is a monitor for STING activation. It is well established that there are alternative ways of STING activation, independent of cGAMP, and there is no reason to suspect that the sensor would not respond to such stimuli. To confirm the contribution of cGAMP to STING activation, the authors would have to mutate the cGAMP-ligand binding domain of the sensor. We suggest that the manuscript's title and content is adjusted to clarify that this tool measures activation of STING, irrespective of the upstream inputs. The manuscript even shows that upon prolonged CPT treatment which leads to autophagy, the biosensor gets activated (Fig. S3F), which is in line with all the literature on the role of STING activation in autophagy.

1) As discussed above, we now referred to the tool as SIRF (STING-IRF3) biosensor and clarify that it functions for STING agonists, including cGAMP.

2) We generated a SIRF biosensor carrying STING^{R238A/Y240A} (Deficient in 2'3'-cGAMP binding, (Gui *et al.*, 2019) and IRF3^{S386A} (Deficient in dimerization, (Jing *et al.*, 2020). We show that this mutant biosensor failed to activate upon cGAMP treatment (Fig. EV2D,E)

2. Throughout the manuscript, it is unclear how many independent experiments are performed in each dataset/figure. Please specify and repeat where necessary.

We now include detailed information on reproducibility in each figure legend.

3. The authors write about the ability of the sensor to report on the dynamics of the activation of the pathway. However this claim is not well substantiated, since they do not characterize the reversibility of the activation. They report that the washout of the STING agonist leads to loss of fluorescence, but this seems to be a slow response and a suboptimal setting. We suggest that this may be better addressed in response to cGAMP treatment. In fact, it is rather concerning/surprising the GFP levels continue to rise for hours on end (up to 10h of STING agonist treatment), while prior literature has demonstrated that once stimulated, the pathway also needs to be switched off, which is driven at least in part by STING degradation. Does the sensor follow this same trend? Are the kinetics of activation/inactivation coinciding with the kinetics of activation/inactivation of endogenous STING? We find that it is important to better characterize the (ir)reversibility of the sensor's activation.

We now provide data of the biosensor inactivation after cGAMP withdrawal (Fig. EV2F), as well as comparisons with phospho-STING and nuclear IRF3. Biosensor signal intensity dropped to 70% after 24 hours of the withdrawal, and displayed a larger inactivation (% of positive cells) compared to nuclear IRF3 (Fig EV2F,G). However, as indicated by the referee, phospho-STING levels are reduced after 2-8h post-activation, without any need of withdrawal (Fig. EV2H), while the biosensor only recovers in 3X that time (Fig. 2E). Together with our previous analyses, these results indicate i) The biosensor reproduces the dynamics of STING activation similarly as phospho-STING and nuclear IRF3 (Fig EV1D-H, EV2B, 3F, EV3A) ii) the biosensor is a good indicator for overall STING-IRF3 activity (i.e. active nuclear IRF3), including during cGAMP withdrawal (Fig. EV2F,G), iii) but does not monitor the fast dynamics phospho-STING termination (Fig. EV2H vs 2E). We now indicate in the manuscript the limitations of the tool in that respect.

4. When characterizing a new system, it is best to have a good reference point. Takaki et al. recently nicely showed IRF3 translocation to be a very sensitive readout for the activity of the pathway. Therefore, we suggest that the figures 1D-E show not only the imaging of the newly developed reporter system but to also show how this compares to IRF3 translocation. We find that further characterization here is also necessary through additional approaches. It would be nice if authors could show that interaction of the tagged STING and IRF3 occurs through a method other than measuring GFP fluorescence. This could be done by PLA or IP but also prevented by brefeldin A. Of course, to assess the contribution of the sensor to the activation, these experiments would be most easily interpretable and designed in the cellular background where no STING or IRF3 is expressed (unlike the HeLa cells the authors establish the sensor in).

We tried both approaches:

1) As discussed above, we now show IRF3 nuclear accumulation in Fig. EV1D,E, and demonstrated that it is blocked by Brefeldin A. We also took advantage of this IF analysis to further validate other results, including for replication stress (Fig. EV4I).

2) We also performed PLA analyses using anti-STING and anti-HA antibodies, similarly as we have previously done for other signalling pathways (Bufe *et al*, 2021). We used HEK293T cells, which do not express endogenous cGAS or STING (Fig. EV1G,H). We noticed PLA signal at the ER in HEK293T biosensor cells, but not in wt HEK293T cells, under basal conditions. These results suggest stochastic interactions and/or unprocessed STING-P2A-IRF3 in the ER, which likely account for the halo signal we report under basal conditions through the manuscript. This basal PLA signal precluded us to perform more detailed analyses upon activation, but we included here the results for your perusal (Referee figure 2).

We believe that the phosphorylation of the STING part of the biosensor (Figure 1C), together with detailed characterization of the tool with agonists, inhibitors (Brefeldin), mutants, and biological inputs, in 3 different cell types -and in parallel to WB and IF analyses of STING and IRF3-, strongly support the relevance of the SIRF biosensor to monitor STING signalling in live cells. Further, we now include an automatic pipeline for SIRF biosensor analyses to avoid biases associated with manual imaging quantification (Figure 2).

Referee Figure 2: Proximity ligation assays (PLA) between anti-STING and anti-HA (IRF3 from the biosensor) in HEK293T cells, which lack endogenous STING or cGAS. White arrows indicate activated biosensor. Note that PLA signal appears around the nuclei under basal conditions in SIRF biosensor cells, possibly due to stochastic interactions in the ER or/and construct processing.

5. To explain the findings in Fig.3 (namely, that many mCherry- cells show activation of the reporter), the authors propose a model of paracrine spreading of signal. What caught our eye is that the highlighted cell in panel A of Fig. 3 which is mCherry- and shows activation of the pathway even before the first transfected responder cells, is found far away from any responding cells. The authors mention that they find no distance-dependency when it comes to this 'bystander' activation while one would expect paracrine signals to act in a distance-dependent manner. We think this model and data needs further investigation, before strong conclusions on paracrine signalling can be drawn.

We thank the referee for this critical suggestion, as our conclusion was indeed affected by the experimental setup of previous Fig 3A.

1) To better understand how paracrine cGAMP signaling interacts with the biosensor, we first transfected HEK293T cells with either mCherry (Control) or cGAS-mCherry. Given that HEK293T cells do not harbour endogenous cGAS (Fig. EV1G,H), mCherry transfected cells are not expected to produce cGAMP, while cGAS-mCherry transfected cells allow for self-triggering of the expressed cGAS by its plasmid. Mixing cGAS-mCherry HEK293T cells with HEK293T SIRF biosensor cells resulted in a wave of activation of the biosensor starting from the cGAS-expressing cells that spread outwards 10-20 μ m/ hour, while control mCherry cells induced no activation (Fig. 3A,B and movies S5, S6). These results could be consistent with cGAMP being transfer through e.g., gap junctions and/or vesicles instead of diluted in the media. Interestingly, cGAS-mCherry HEK293T cells also activated HeLa SIRF biosensor cells in a distance-dependent manner, albeit reaching a lower radius and speed (< 5m /

hour) (Fig. 3C,D). These results may indicate differences in cell junctions and other paracrine methods, activation dynamics, as well as different cell size and spacing between HEK293T and HeLa cells, and highlight the relevance of the biosensor to monitor cGAMP spreading across a cell population.

2) To avoid problems associated with the late expression of mCherry in the previous experiments of Figure 3, and following the advice of Referee #3, we directly transfected fluorescently labelled DNA (Plasmid+TOTO3, Fig. 3E)) in to HeLa biosensor cells. We found that activation of the biosensor in TOTO3 positive preceded the activation in TOTO3 negative cells in around 3 hours (Fig. 3E-G).

3) We believe that previous results related to plasmid transfection in HeLa cells reflected a salt-and-pepper transfection efficiency and expression delays, which prevented us to separate unequivocally paracrine vs intrinsic signalling. In that line, we noticed that previous treatments with ABCC1i reduced the number of transfected cells (Referee Figure 3), which likely accounted for the reduced activation of the biosensor in the HeLa population (Previous Figure 3). As such, we decided to switch the intercellular analyses with the experiments described above.

Referee Figure 3: *ABCC1i reduces the proportion of transfected cells (mCherry) which likely accounts for a reduction of biosensor activity in bystander HeLa cells.*

6. Further, we find the result in Fig. 5J to be very surprising, as it shows that the response is much higher in the cells where the direct stimulus is not present (paracrine stimulation is higher than the direct transfection effects). The authors try to show that the biosensor is responsive to multiple sources of stimulation. However, with such unexpected results in this experiment, we think it would be important to do a more careful characterization of the system's responsiveness to exogenous cGAMP (limited data shown in Fig. 1I), similar to the characterizations in Fig. 5.

We believed the referee refers to previous Figure 3F, which was unfortunately mislabelled and showed % of cells mCherry positive and negative in the population, instead the % of each population with active biosensor. We apologise for the mistake.

Nevertheless, in 3 independent experiments with HSV-1 we found that only 10% of HSV-1 infected HeLa cells (mCherry+) display activation of the biosensor (Fig. 4A-C) compared to 80% of cells transfected with dsDNA (Fig 3G, (TOTO3+) while neighbouring cells (Negative for either mCherry (Infection experiment) or TOTO3 (Transfection experiment)) showed in both cases between 15-20% activation rates. We hypothesise that a reduced response of the biosensor in HSV-1 infected cells might be associated with previously reported cell intrinsic inhibition of the innate immune response by this virus (Deschamps & Kalamvoki, 2017; Drayman *et al*, 2019; Hare *et al*, 2020; Sun *et al*, 2015), which is not produced by e.g., dsDNA (Figure 3) or mtDNA (Figure 5). Further, we now show that inhibition of viral replication using acyclovir had limited effect on the biosensor response in infected cells, but largely blocked activation in neighbouring cells (Fig. 4D).

7. The data on the activation of the sensor gained in response to more physiological stimulation would be beneficial. Most of the experiments show that the system responds well to a very strong STING agonist (again, with limited data following stimulation with cGAMP shown in Fig. 1I).

In addition to the discussed experiments of paracrine cGAMP upon dsDNA (Fig. 3-4), we now include a titration of cGAMP fully analysed with a new automatic pipeline in (Fig. 2G,H, and See below). These new results further add to our other physiological readouts (mtDNA in Fig. 5 and TREG1 deficiency in Fig. EV5). Of note i) our biosensor results upon HSV-1 infection confirm previous reports on viral inhibition of STING signalling (Deschamps & Kalamvoki, 2017; Drayman *et al.*, 2019; Hare *et al.*, 2020; Sun *et al.*, 2015), while ii) our results monitoring micronuclei (i.e. no activation of the biosensor) are backed up by WB and IF analyses of other STING components (Fig. 6 and EV4).

Taken together we believe that with the current results, we demonstrate that the biosensor reliably monitors STING signalling upon a broad range of physiological stimuli.

Minor comments

1. As previously mentioned in one of our major comments, it is unclear which part of the activation of the pathway can be attributed to the biosensor and which is based on the endogenously-expressed STING and IRF3. We find that the WB in Fig.1C is mislabeled, as the authors indicate that they detect GFP-tagged portions of STING when both endogenous and 'biosensor' pools of STING would be detected by a p-STING antibody. It is unclear what the 2 bands of the pSTING part of the blot represent. The authors need to clarify which band is which. Also, we suggest blotting for IRF3 levels in Fig. 1C. This would give a better description of the reporter system.

We now clarify this point with new WBs analyses in Figure 1C, which show the protein levels and activation of either endogenous or stable transfected STING. Both stably transfected IRF3 and STING are expressed in higher levels compared to their endogenous counterparts, but present similar activation levels for TBK1 (Fig. EV1B). Of note HEK293T biosensor cells which do not have endogenous STING (Fig. EV1G,H, 3A,B), functions similarly as the HeLa biosensor cells.

Mechanistically, we expect that both endogenous and transgenic STING signalling components (when present) should be interlinked in the complexes.

2. Figure 3H, please show that Influenza A sequence is present in cells.

We now show in the Figures 4A and EV3B the successful infection of Influenza A both by InCuCyte imaging and SC35M-specific qPCR.

3. Figure S3G, please show TREX1 levels on western blot upon siTREX1.

We now show the KD efficiency in Figure EV5D.

4. Please describe in materials and methods how the treatments with cGAMP are performed. How is the mock sample treated?

To avoid perturbations and heterogeneity caused by electroporation, which is a method used in several previous publications, we treated cells directly with cGAMP or vehicle (Hepes). We now described in the methods the automatic CellProfiler pipeline utilised for those analyses.

5. On page 7 of the manuscript the authors write that micronuclei not activating the pathway is unexpected. Please refer to Flynn et al, 2021 here. This is the first paper (to our knowledge) to report that micronuclei are not (strong) activators of STING.

We now cite this manuscript together with the other recent studies on the matter in the introduction, results and at the end of the discussion.

We would like to point out the Flynn et al do not measure active STING activity and but rely on cGAS and interferon/cytokine production, which we (Fig. EV4) and contemporaneous studies (Takaki *et al.*, 2024) demonstrate that can be independent of cGAS-STING-IRF3 signalling. In the light of our results, we think it would be important to revisit whether chromatin bridges do activate STING or function in IFN signalling through other mechanism.

6. Please show that HEK293T cells do not express cGAS and STING. (fig. S2C)

We show now WBs analyses in supplementary Figure EV1H.

7. In Fig. 3A at timepoint 20h, the cells appear to be very unhappy. It is difficult to see since the only channel shown is GFP. In any case, we think it is important to show that the stimulation of the biosensor is not toxic.

We did not detect cell death upon stimulation with e.g., diABZI or cGAMP. Some experiments such as high levels of DNA transfection can lead to residual cell death after more than 20 hours both in wt or biosensor cells. We now show a long video of continuous recording of the biosensor and include the DIC (movie S10). Furthermore, we now indicate that we had passaged HeLa biosensor cells after biosensor activation without any complications in both methods and main text.

8. Fig. 4C does not have a QvD only condition.

We did not run QvD alone in those set of experiments as we had previously tested that had no effect. We included it as referee figure 4 for your perusal.

Referee Figure 4: QvD alone does not activate the biosensor in HeLa cells

Referee #3:

In this work, Smarduch et al. generate a novel biosensor for cGAMP detection in cells using a split GFP system tagged to STING and IRF3. The authors show that this system, when stably expressed in cells, can be activated by STING agonists (synthetic and cGAMP), viruses, DNA (with cGAS activation), mitochondrial DNA upon apoptosis, but not in the context of micronuclei. Overall the authors provide convincing evidence that this system would be extremely useful to easily monitor the live propagation of cGAMP between cells in tissues. This tool should represent a very useful solution to the monitoring of cGAMP dynamics, however a few points need to be addressed to broaden its utility.

Thank you.

Major points:

1- The authors rely heavily on two aspects of their biosensors; on one hand, the biosensor forms intracellular aggregates/punctuates, as previously reported; On another these have increased fluorescence compared to background fluorescence. What is not very clear is what is used in which figures and how easily this is assessed. There is little information about how many cells were counted for the MFI (200 seems to be the cut of for punctate positive cells) - but the methods suggest a lot of this was done manually.

The authors seem to switch between one and the other in their figures - which is a bit confusing (although this reviewer understands that the authors are trying to show different aspects of the system). If the cGAMP GFP intensity can indeed be accurately measured and is increased upon stimulation, then the assay should also be validated on flow cytometry for a true quantification of fluorescence at the single cell level (which would be unbiased and more accurate than the user selected cells measured by microscopy). The abstract suggests that the biosensor can be used for plate reader (presumably whole well assay in fluorescent plate reader) - this should also be demonstrated somewhere. Ultimately, the authors need to better support the single cell level approach using unbiased analyses.

In the original version we analysed the biosensor either by manual quantification of GFP intensity in the foci (most analyses) or in a Incucyte plate reader (Former Figures 3H-J and Figure 4 on infection and apoptosis).

We agree with the referee that unbiased and scalable approaches are preferred and addressed this question in different ways:

1- Confocal imaging + CellProfiler: We implemented a fully automatized segmentation and quantification pipeline using CellProfiler in SIRF biosensor H2B-mCherry HeLa cells that

calculates the biosensor dynamics at single cell level (activation, number of punctae, average intensity, integrated intensity) from each all the imaged cells across time (Figure 2D,E). We now showcase the efficacy of this approach for unbiased analyses of the biosensor for different conditions, including cGAMP (Figure 2G,H), and refined to further characterise automatically signalling in different subgroups, including dsDNA transfected vs non-transfected cells (Figure 3F,G). The combination of the biosensor with the CellProfiler approach (We include the CellProfiler code as supplementary material) will strongly facilitate researchers monitoring STING activity at single cell levels across a heterogeneous population and time. We provide the CellProfiler data for this approach as “Biosensor integrated intensity (RU)” “Biosensor activated cells (%)”, and explain the process in the methods.

2- Incubator imaging using IncuCyte: Our co-authors working in apoptosis and viral infection rely on IncuCyte plate readers for their studies. This allowed us to demonstrate the utility of our tool beyond classical confocal imaging. We now show analyses with IncuCyte plate reader for most of the perturbations followed up with automatic quantification of GFP signal provided by the equipment (See methods). These analyses largely confirmed our previous manual analyses albeit with slightly lower sensitivity compared to confocal imaging (See Figure 2), and demonstrated the efficiency of this type of plate reader, especially for population analyses of the biosensor. This approach is suitable for both live cell imaging and IF, and can be easily scale up for high-throughput analyses. Given that this is not a standard plate reader (See below), we now clearly indicate that we refer to this instrument.

3- Manual analyses: We now leave manual analyses for the proof-of-principle experiments of Fig. 1-2, as well for studies of e.g., micronuclei (Fig. 6), which require expert monitoring of the different subpopulations.

4- We examined the biosensor in a regular plate reader instead of an IncuCyte. We could not detect differences in activation possibly due to background signal and/or low sensitive of this method.

5- We also performed FACS analyses and only detected minor differences using this approach. Given the basal signal of the biosensor in the ER (Fig. 1), we believe that a more sophisticated approach such as Imaging flow cytometry (IFC) should be require to characterize the biosensor through cytometry.

Taken together, our previous and new analyses support three robust type of analyses for the biosensor: i) Manual characterization of median intensity in foci after live cell imaging or IF, ii) automatized segmentation and analysis of biosensor cells using CellProfiler and a nuclear marker after live cell imaging or IF, and iii) Automatic population analyses using IncuCyte (which can be complemented with manual analyses for single cell characterization). Beyond describing the 3 approaches in detail in the manuscript, we now indicate the rationale of using either approach along the manuscript and clearly indicate them in the figure legends and methods.

2- One of the key aspects of this research is the capacity afforded by this technology to assess the propagation of cGAMP within tissue. The authors touched on this but do not address the capacity of cGAMP to propagate through gap junctions. Since HEK 293T cells form CX43 dependent gap-junctions, it would be easy to confirm this with minimal experiments. For example, co-culture of a few HEK 293T cells overexpressing cGAS-GFP

(which make cGAMP) added on top of a confluent layer of HEK293T cells with the STING-IRF3 biosensor would allow to measure activation of the signal as it progresses through the cell monolayer.

We thank the referee for this important suggestion.

As discussed above (See Referee #2, question #5), we performed this experiment and analysed cells by either live imaging (Fig. 3A-D and movies S5, S6). We detected a fast and wave-like spread of biosensor activation in HEK293T cells, and slow and limited spread in HeLa cells. We now discuss these findings in the manuscript, comment on the relevance of cellular junctions, as well as the utility of our tool to examine cGAMP transfer.

3- The authors propose that micronuclei do not activate their biosensor. Because the micronuclei function/sensing would be expected to be very different between cell types it is important to tone-down the fact that this may be the case in their model - i.e. HeLa cells - but maybe not in other models. One point which should be definitely be mentioned is the case of non-canonical STING signaling, which does not depend on cGAS and thus would not be able to recruit the biosensor. In this regard, it would be very useful to create stable HaCaT cells expressing this biosensor and show that while diABZI activate the biosensor, DNA damage with CPT/ does not (see Dunphy et al., Al-Asmari et al.), although it activates production of IL-6. This would complete the characterisation of this reporter and nuance the findings on sensing of DNA damage products presented here.

We clarify now that our results are limited to HeLa and HEK293T cells (See Fig. EV4L). However, contemporaneous studies also show that micronuclei are poor activators of STING in HeLa Kyoto, SiHa, WI-38, HaCaT HUVEC, T24, and HCT116 cells (Sato & Hayashi, 2024; Takaki et al., 2024).

We tried to generate HaCat biosensor cells but i) the transfection efficiency was really low and the cells grow slowly thereby complicating the generation of positive clones. Of note, we do not detect activation of STING signalling upon CPT or aphidicolin in HeLa cells either, unless is associated with cell death / autophagy (Fig. 6 and EV4). In the light of our results on cytokine expression upon aphidicolin and reversine, including in cGAS lacking cells such as HEK293T cells (Fig EV4L), as well as similar results in two contemporaneous studies (Sato & Hayashi, 2024; Takaki et al., 2024), we believed that it would be important revisiting previous studies linking DNA damage to STING signalling, specially if they were based on either i) cGAS recruitment or ii) IFN/cytokine expression data.

4- The sensitivity of the assay should be determined using fluorescent labelled dsDNA (e.g. ISD70), transfected in cells and biosensor activity in transfected cells measured with time. Unlike the DNA transfection used here (noting that the authors used SV40-mcherry which may increase the amount of cytosolic vector if the vector has an SV40 ori), transfection of defined amount of labelled synthetic DNA will allow to define a molecular concentration of cytosolic DNA sensed. This should be compared to the induction of IRF3 targets by RTqPCR to similar amount of transfected DNA in HeLa cells.

We previously used mCherry with only the NLS from SV40, not the SV40 ori. Nevertheless, as suggested by the referee, and above discussed with referee #2, our previous approach had a number of caveats (See response to Referee #2, question #5).

We had problems obtaining ISD-derived reagents with the required fluorophore, and used instead Thiazole Red Homodimer (TOTO3) labelling followed by DNA-fluorophore precipitation, and transfection. We show i) DNA concentration-dependent activation of the biosensor upon transfection reaching similar levels as full activation with diABZI, and ii) that activation of the biosensor in TOTO3 positive preceded the activation in TOTO3 negative cells in around 3 hours (Fig. 3E-G).

We also show qPCR analyses of HeLa cells treated with the same concentration of DNA (Fig EV3A). The biosensor shows a comparable sensitivity to *CXCL10* expression albeit faster dynamics (Fig. EV3A vs 3F), consistent with an upstream response. Of note the biosensor was more sensitivity than *IFNB* and *IL6* expression (See blue line in EV3A vs 3F).

Minor points:

- the text refers to Figure S1C but this should be S2 (in results section).

Changed

-the abbreviation for interferon is IFN not INF (which is used throughout the paper)

We apologize for the error. We corrected it in the figures and text.

- Why do they use SV40-mcherry? - noting that SV40 has an impact on sensitization of cells to cGAS sensing by DNA damage (Pepin et al, mBio 2018).

We only used the NLS of SV40 to ensure nuclear localization. We now used fluorescently labelled DNA, as per referee's suggestion.

References

- Bufe A, Garcia Del Arco A, Hennecke M, de Jaime-Soguero A, Ostermaier M, Lin YC, Ciprianidis A, Hattmer J, Engel U, Beli P *et al* (2021) Wnt signaling recruits KIF2A to the spindle to ensure chromosome congression and alignment during mitosis. *Proc Natl Acad Sci U S A* 118
- Deng L, Liang H, Xu M, Yang X, Burnette B, Arina A, Li XD, Mauceri H, Beckett M, Darga T *et al* (2014) STING-Dependent Cytosolic DNA Sensing Promotes Radiation-Induced Type I Interferon-Dependent Antitumor Immunity in Immunogenic Tumors. *Immunity* 41: 843-852
- Deschamps T, Kalamvoki M (2017) Evasion of the STING DNA-Sensing Pathway by VP11/12 of Herpes Simplex Virus 1. *J Virol* 91
- Drayman N, Patel P, Vistain L, Tay S (2019) HSV-1 single-cell analysis reveals the activation of anti-viral and developmental programs in distinct sub-populations. *Elife* 8
- En A, Bogireddi H, Thomas B, Stutzman AV, Ikegami S, LaForest B, Almakki O, Pytel P, Moskowitz IP, Ikegami K (2024) Pervasive nuclear envelope ruptures precede ECM signaling and disease onset without activating cGAS-STING in Lamin-cardiomyopathy mice. *Cell Rep* 43: 114284
- Flynn PJ, Koch PD, Mitchison TJ (2021) Chromatin bridges, not micronuclei, activate cGAS after drug-induced mitotic errors in human cells. *Proc Natl Acad Sci U S A* 118
- Gui X, Yang H, Li T, Tan X, Shi P, Li M, Du F, Chen ZJ (2019) Autophagy induction via STING trafficking is a primordial function of the cGAS pathway. *Nature* 567: 262-266

- Harding SM, Benci JL, Irianto J, Discher DE, Minn AJ, Greenberg RA (2017) Mitotic progression following DNA damage enables pattern recognition within micronuclei. *Nature* 548: 466-470
- Hare DN, Baid K, Dvorkin-Gheva A, Mossman KL (2020) Virus-Intrinsic Differences and Heterogeneous IRF3 Activation Influence IFN-Independent Antiviral Protection. *iScience* 23: 101864
- Hong C, Schubert M, Tjihuis AE, Requesens M, Roorda M, van den Brink A, Ruiz LA, Bakker PL, van der Sluis T, Pieters W *et al* (2022) cGAS-STING drives the IL-6-dependent survival of chromosomally instable cancers. *Nature* 607: 366-373
- Jing T, Zhao B, Xu P, Gao X, Chi L, Han H, Sankaran B, Li P (2020) The Structural Basis of IRF-3 Activation upon Phosphorylation. *J Immunol* 205: 1886-1896
- Li D, Wu M (2021) Pattern recognition receptors in health and diseases. *Signal Transduct Target Ther* 6: 291
- Marinello J, Arleo A, Russo M, Delcuratolo M, Ciccarelli F, Pommier Y, Capranico G (2022) Topoisomerase I poison-triggered immune gene activation is markedly reduced in human small-cell lung cancers by impairment of the cGAS/STING pathway. *Br J Cancer* 127: 1214-1225
- Martin S, Scorzoni S, Cordone S, Mazzagatti A, Beznoussenko GV, Gunn AL, Di Bona M, Eliezer Y, Leor G, Ben-Yishay T *et al* (2024) A p62-dependent rheostat dictates micronuclei catastrophe and chromosome rearrangements. *Science* 385: eadj7446
- Mohr L, Toufektchan E, von Morgen P, Chu K, Kapoor A, Maciejowski J (2021) ER-directed TREX1 limits cGAS activation at micronuclei. *Mol Cell* 81: 724-738 e729
- Nader GPF, Aguera-Gonzalez S, Routet F, Gratia M, Maurin M, Cancila V, Cadart C, Palamidessi A, Ramos RN, San Roman M *et al* (2021) Compromised nuclear envelope integrity drives TREX1-dependent DNA damage and tumor cell invasion. *Cell* 184: 5230-5246 e5222
- Sato Y, Hayashi MT (2024) Micronucleus is not a potent inducer of the cGAS/STING pathway. *Life Sci Alliance* 7
- Sun C, Schattgen SA, Pisitkun P, Jorgensen JP, Hilterbrand AT, Wang LJ, West JA, Hansen K, Horan KA, Jakobsen MR *et al* (2015) Evasion of innate cytosolic DNA sensing by a gammaherpesvirus facilitates establishment of latent infection. *J Immunol* 194: 1819-1831
- Takaki T, Millar R, Hiley CT, Boulton SJ (2024) Micronuclei induced by radiation, replication stress, or chromosome segregation errors do not activate cGAS-STING. *Mol Cell* 84: 2203-2213 e2205

Dear Sergio,

Thank you again for submitting your revised manuscript (EMBOJ-2024-118005R) to The EMBO Journal for our consideration. It has now been seen by the three referees who had previously assessed the original version of your manuscript, and we have received their comments, which are included below.

I am glad to say that all three referees recognize that the revised manuscript is significantly improved, find their previously raised concerns sufficiently addressed, and now support publication of the manuscript in our journal. In light of this input, I am happy to inform you that your manuscript has been in principle accepted for publication in The EMBO Journal. Congratulations on an excellent work and thank you for your comprehensive responses to the referees' comments!

There are some minor changes and corrections that must be made in a final version of the manuscript before we can proceed with its publication:

- Since the focus of the work is largely on the development and validation of a new reporter/biosensor, we would recommend publishing this manuscript as a "Method" article rather than a research article. We would therefore kindly ask you to choose the Manuscript Type "Method" when you submit your final version of your revised manuscript to our manuscript handling system.

- If there is any relevant literature published during revision and review of your manuscript that is not discussed/cited in your manuscript yet, we would kindly ask you to mention and cite these studies in the final version of your manuscript.

- All funding sources should be listed in the Acknowledgements section of your manuscript and also entered in our manuscript handling system (eJP) during resubmission. Information related to the grant "Heidelberg University through the Excellence Initiative - Explorer programme" is currently missing from eJP.

- Please note that you may list up to 5 relevant keywords after the Abstract of your revised manuscript (you currently have 7).

- Please change the heading "Ethics declarations" to "Disclosure and competing interests statement".

- The author contributions statement should be removed from the manuscript file. Instead, we use CRediT to specify the contributions of each author in the journal submission system. Please feel free to use the free text box to provide more detailed descriptions during submission. See also our guide to authors for more information:

<https://www.embopress.org/page/journal/14602075/authorguide#authorshipguidelines>.

- We noticed that callouts for Figure 6J are missing; please make sure that all Figure panels are called out in your revised manuscript.

- Please note that the sections in the last column of your Author Checklist should be specified only for positive response (not if the response is "Not Applicable"). In the last column, please provide only the sections of the manuscript where the relevant information can be found. Please revise your Author Checklist and upload the corrected version with your resubmission.

- Please delete the Instructions from the beginning of your Reagents and Tools Table, and upload the revised Table (as a "Reagent Table").

- Please note that EMBO press papers are accompanied online by:

- A) a short (2 sentences) summary of the findings and their significance,

- B) 2-5 short bullet points highlighting the key results, and

- C) a synopsis image in .jpg or .png format that is exactly 550 pixels wide and 300-600 pixels high (the height is variable). Please note that the text needs to be legible at the final size.

Please upload this information along with your revised manuscript (the text for A and B should be provided in a separate Word file).

- During our routine pre-acceptance checks, our data editors have raised the following queries regarding figures, data, and legends. Please make sure that all requests below are completely addressed in the final version of your manuscript:

1. Please define the annotated p values *** as well as provide the exact p-values for the same in the legend of Figure 4d; EV 5f; as appropriate.

2. Please note that the exact p values are not provided in the legends of Figures 1f; 6d-f; EV 2e; EV 4i-j; EV 5b-c.

3. Please indicate the statistical test used for data analysis in the legends of Figures 4d; EV 5f.

4. Please note that in Figures 6d-f; EV 2e; EV 5b-c; there is a mismatch between the annotated p values in the Figure legend and the annotated p values in the Figure file that should be corrected.
5. Please note that information related to "n" is missing in the legends of Figures EV 1c, g; EV 2j; EV 3a; EV 5b-c, f.
6. Please note that the error bars are not defined in the legends of Figures 6i; EV 1c, g; EV 2j; EV 3a; EV 4d, i-l; EV 5b-c, f.
7. Please note that the scale bar is missing for Figure EV 4a.
8. Please note that the scale bar needs to be defined for Figures 5d-e.
9. Please note that the white arrows/arrowheads are not defined in the legend of Figure 6c, h; EV 2a. This needs to be rectified.

- The movie files should be renamed to "Movie EV1-EV10" with the corresponding callouts updated accordingly in the manuscript file. Their legends should be removed from the manuscript file and instead provided in separate text files zipped together with their corresponding movies.

- The section "ANNEXED CODE" should be removed from the manuscript file. All new code generated and used in this study should be deposited in an appropriate repository and listed in the "Data availability" statement (please provide for each code the repository, identified, and specific URL).

- Please also add the link to your imaging data in the Biolineage Archive in your SD checklist and the Data availability statement of the manuscript.

- The section order should be corrected as follows: Title page - Abstract & Keywords - Introduction - Results - Discussion - Methods - Data Availability - Acknowledgements - Disclosure and Competing Interests Statement - References - Figure Legends - (main manuscript Tables, if there are any) - Expanded View Figure Legends.

Please also note that as part of the EMBO publications' Transparent Editorial Process, The EMBO Journal publishes online a Peer Review File along with each accepted manuscript. This File will be published in conjunction with your paper and will include the referee reports, your point-by-point response and all pertinent correspondence relating to the manuscript. You can opt out of this by letting the editorial office know (contact@embojournal.org). If you do opt out, the Peer Review File link will point to the following statement: "No Peer Review File is available with this article, as the authors have chosen not to make the review process public in this case."

We look forward to seeing a final version of your manuscript as soon as possible. Please let us know if you have any questions and use this link to submit your revision: <https://emboj.msubmit.net/cgi-bin/main.plex>.

Best wishes,

Ioannis

Referee #1:

This reviewer appreciates the insightful responses and additional experiments during the revision. The manuscript has been much improved, and I believe that it is now ready for publication.

Referee #2:

As we had already written a detailed report in reference to the first submission, here we would like to only refer to the changes made to improve the manuscript.

We find that the changes the authors made improved the manuscript greatly and give more confidence to their findings and the newly established reporter system. All our comments were addressed to a satisfactory level. We find that the focus of the manuscript was to establish and validate a new reporter, and not so much to unravel new biology. However, the additional experiments performed on paracrine signaling have added value for the reader.

Referee #3:

The authors have addressed all my concerns and have now added key experiments/clarification for the use of their novel biosensor; this makes this work a substantial contribution to the field.

All editorial and formatting issues were resolved by the authors.

Dear Sergio,

Happy New Year and congratulations on an excellent manuscript! I am very pleased to inform you that it has been accepted for publication in The EMBO Journal. Thank you for your comprehensive responses to the referees' comments and for addressing all editorial and formatting requests.

If you have any questions, please do not hesitate to contact the Editorial Office. Thank you for your contribution to The EMBO Journal. Working with you has been a pleasure!

Best wishes,

Ioannis
